# From Statics to Dynamics:
# Physics-Aware Image Editing with Latent Transition Priors

Liangbing Zhao[1]  Le Zhuo[2][3]  Sayak Paul[4]  Hongsheng Li[2]  Mohamed Elhoseiny[1]

## Abstract

Instruction-based image editing has achieved remarkable success in semantic alignment, yet state-of-the-art models frequently fail to render physically plausible results when editing involves complex causal dynamics, such as refraction or material deformation. We attribute this limitation to the dominant paradigm that treats editing as a discrete mapping between image pairs, which provides only boundary conditions and leaves transition dynamics underspecified. To address this, we reformulate physics-aware editing as predictive physical state transitions and introduce *PhysicTran38K*, a large-scale video-based dataset comprising 38K transition trajectories across five physical domains, constructed via a two-stage filtering and constraint-aware annotation pipeline. Building on this supervision, we propose *PhysicEdit*, an end-to-end framework equipped with a textual-visual dual-thinking mechanism. It combines a frozen Qwen2.5-VL for physically grounded reasoning with learnable transition queries that provide timestep-adaptive visual guidance to a diffusion backbone. Experiments show that PhysicEdit improves over Qwen-Image-Edit by 5.9% in physical realism and 10.1% in knowledge-grounded editing, setting a new state-of-the-art for open-source methods, while remaining competitive with leading proprietary models. All code, checkpoints, and datasets are available at https://liangbingzhao.github.io/statics2dynamics/.

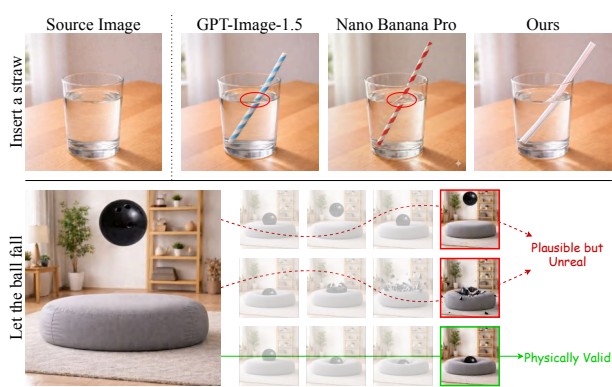

Figure 1. **Bridging semantic alignment and physical plausibility.** (Top) Despite high semantic fidelity, existing editing models frequently violate physical principles. (Bottom) Traditional image editing treats editing as a black box, learning a discrete mapping with underspecified constraints. Our approach reformulates editing as a **Physical State Transition**, leveraging continuous dynamics to constrain the state transition space from unreal hallucinations to physically valid trajectories.

## 1. Introduction

Instruction-based image editing, which aims to generate a new image following the given instruction, enables users to do visual creation much more easily. As user demands evolve from simple style transfer or object replacement to more complex scenarios involving hypothetical or counterfactual changes, the focus has shifted toward *reasoning-based* editing (Huang et al., 2024; He et al., 2025; Zhuo et al., 2025b). This demand has contributed to the emergence of unified multi-modal models (UMMs), such as Bagel (Deng et al., 2025) and Nano Banana (Google, 2025), which leverage the inherent textual reasoning capabilities of Multimodal Large Language Model (MLLMs) to bridge the gap between abstract user intent and visual execution. While these unified frameworks have established a new backbone for image editing, their reasoning modules operate primarily at the semantic level rather than physical causality. Consequently, despite high semantic fidelity, state-of-the-art models frequently hallucinate artifacts that violate fundamental physical principles, as they prioritize object matching over physical plausibility.

This drawback becomes particularly apparent in scenarios

[1]KAUST [2]CUHK MMLab [3]Krea AI [4]Huggingface. Correspondence to: Hongsheng Li <hsli@ee.cuhk.edu.hk>, Mohamed Elhoseiny <mohamed.elhoseiny@kaust.edu.sa>.

*Proceedings of the 43rd International Conference on Machine Learning*, Seoul, South Korea. PMLR 306, 2026. Copyright 2026 by the author(s).

governed by strict physical interactions. Consider a common scenario: inserting a straw into a transparent glass of water. As shown in Figure 1, while existing models can correctly identify the object ("straw") and the location ("in the glass of water"), they frequently fail to render the *optical refraction* phenomenon, where the straw should appear disjointed or bent at the water surface. Instead, they tend to generate a naive straw's position, maintaining geometric rigidity while violating optical physical laws. This discrepancy reveals a fundamental limitation: current models maximize semantic alignment at the cost of physical plausibility.

To bridge this gap, we argue that the next frontier in visual creation lies in physics-aware image editing: a paradigm where generated content must rigidly adhere to the causal rules of the physical world. To achieve this goal, we propose a fundamental shift in problem formulation: the editing process should not be viewed as a static mapping between independent images, but rather as a predictive physical state transition. Under this formulation, the source image represents an initial state, and the edit instruction specifies an external interaction or trigger that drives the scene toward a subsequent state under physical laws.

A central challenge is that standard paired images supervision in image editing provides only boundary conditions, leaving the transition itself underspecified. By contrast, temporal sequences show intermediate evidence of how states evolve, making video a natural source of supervision for learning transition priors. Therefore, we construct PhysicTran38K, a large-scale video-based dataset tailored to physical state transitions. Unlike existing benchmarks that predominantly emphasize semantic operations (e.g., add/remove/replace), PhysicTran38K is organized around interaction-driven triggers and law-governed transitions. We design hierarchical physics categories covering five major physical domains, 16 intermediate sub-domains, and spanning 46 distinct transition types. By employing a two-stage filtering pipeline, we finally get and annotate 38,000 high-quality transition data, providing explicit supervision for how physical states evolve over time.

However, leveraging video data introduces a practical mismatch: while videos supervise training, intermediate states are unavailable during inference. We address this train-test discrepancy by introducing PhysicEdit, a physics-aware editing framework built on Qwen-Image-Edit (Wu et al., 2025a) that learns from video trajectories while remaining compatible with the single-image inference workflow. We propose a textual-visual dual-thinking mechanism that decouples physical understanding into two branches. First, a physically-grounded reasoning branch uses a frozen Qwen2.5-VL-7B (Bai et al., 2025) to produce structured physical constraints as textual context. Second, an implicit visual thinking branch introduces learnable transition queries that are trained to learn transition priors from video. Concretely,

intermediate keyframes provide supervision through two complementary encoders (DINOv2 (Oquab et al., 2023) for structural semantics and a VAE (Wu et al., 2025a) for fine-grained appearance), and the transition queries are aligned to these structure- and texture-level targets via dual projection heads. Finally, we align this guidance with diffusion's coarse-to-fine generation through a timestep-aware modulation strategy that emphasizes structure at high noise and texture details at low noise.

In summary, our contributions are as follows:

- We propose PhysicEdit, an end-to-end physics-aware editing framework with a textual-visual dual-thinking mechanism. It combines physically-grounded reasoning with implicit visual thinking to leverage transition priors learned from videos, enabling physically faithful edits.

- We construct PhysicTran38K, a large-scale video-based dataset of approximately 38k video-instruction pairs organized by hierarchical physics categories.

- Extensive experiments demonstrate that PhysicEdit achieves state-of-the-art performance among evaluated open-source models, while performing comparably to leading proprietary models, establishing a strong baseline for physics-aware image editing.

## 2. Related Works

Benefit from powerful diffusion models (Ho et al., 2020; Song et al., 2021; BAKR et al.) in generating high-fidelity images, instruction-based image editing has made rapid progress. Early diffusion-based editors (Hertz et al., 2022; Zhao et al., 2023) typically manipulate cross-attention or invert latents to trade off content preservation and edit strength, but often lack fine-grained controllability under complex structural changes. This motivates instruction-tuned approaches (Brooks et al., 2023; Labs, 2024; Wei et al., 2024; Zhuo et al., 2025a;b) and, more recently, unified multimodal models for generalized instruction alignment. A representative example is Qwen-Image-Edit (Wu et al., 2025a), which conditions an MMDiT (Esser et al., 2024) on frozen Qwen2.5-VL (Bai et al., 2025) multimodal representations to replace standard text embeddings. In parallel, video data has been explored as a prior for improving editing. Earlier works (Deng et al., 2025; Xiao et al., 2024; Chen et al., 2025) construct training pairs from video keyframes to enhance consistency, whereas recent efforts (Wu et al., 2025c; Rotstein et al., 2025) shift toward generative reasoning. Notably, the concurrent work ChronoEdit (Wu et al., 2025c) explicitly synthesizes intermediate frames as reasoning steps but incurs substantial computation and potential error accumulation. In contrast, we adopt an implicit paradigm that distills physical state transition priors into compact latent queries,

enabling efficient feature-space dynamics simulation while inheriting physical fidelity from video data. A more detailed discussion of related work is provided in Appendix C.

## 3. Method

### 3.1. Problem Formulation

To bridge the gap between semantic alignment and physical fidelity, we first formalize the task of physics-aware image editing. Conventionally, instruction-based image editing is modeled as a *discrete* conditional mapping. Given a source image $I_{src}$ and an editing instruction $T_{edit}$, the model approximates a function $\mathcal{F}$:

$$I_{tgt} = \mathcal{F}(I_{src}, T_{edit}), \tag{1}$$

where the goal is to generate a target image $I_{tgt}$ that follows $T_{edit}$ while preserving relevant content from $I_{src}$. While effective for semantic edits (e.g., changing a dog to a cat), this formulation treats the transformation as a black-box pixel update and does not explicitly model the underlying dynamics that govern how the scene should evolve.

In this work, we treat physics-aware editing not as a static mapping, but as a **Physical State Transition**. Let $I_{src}$ represent the initial physical state $S_0$ of a scene, and $T_{edit}$ represent an external force or interaction trigger ( *e.g.*, "drop the glass"). The editing process is effectively a simulation of the time-evolution of the system under physical laws $\Omega$ ( *e.g.*, gravity, fluid dynamics):

$$S_{final} = S_0 + \int_0^\tau \Phi(S_t, T_{edit}; \Omega)\, dt, \tag{2}$$

where $\Phi$ denotes the state transition dynamics and $\tau$ is the duration of the interaction. The desired target image $I_{tgt}$ is the visual outcome of the accumulated state $S_{final}$.

The fundamental challenge arises because standard image editing dataset provide only the boundary conditions, *i.e.*, $(I_{src}, I_{tgt})$, leaving the transition dynamics $\Phi$ completely underspecified. As illustrated in Figure 1, for a given pair $(I_{src}, T_{edit})$, there may exist multiple visually plausible endpoints, yet only a subset results from a valid physical trajectory under $\Omega$. Without constraints on the intermediate integral, models tend to violate specific physical laws that satisfy the instruction. We therefore leverage videos as supervision, since they expose intermediate states and directly constrain how states evolve over time, motivating our construction of PhysicTran38K (Section 3.2). During inference, however, intermediate states are unavailable, so we propose PhysicEdit (Section 3.3) to distill these transition priors from videos into a latent representation, effectively guiding the generation along a physically valid trajectory.

### 3.2. Physics-Driven Data Construction

We construct PhysicTran38K, a video-based dataset for physics-aware image editing, by casting editing as *physical state transitions*. As illustrated in Figure 2, our data construction pipeline is designed to transform hierarchical physics categories into high-quality video data through three stages: structured generation with a video generation model, camera movement and principle-driven verification, and constraint-aware reasoning generation. We provide all the system prompts in Appendix H.

**Hierarchical Physics Categories.** We begin by establishing a comprehensive physics taxonomy to ensure broad coverage of physical dynamics. As shown in Figure 2(a), we organize physical laws into five primary state domains: *Mechanical*, *Thermal*, *Material*, *Optical*, and *Biological*. Under these domains, we define 16 intermediate sub-domains and 46 transition types (e.g., *refraction*, *melting*, *germination*), which serve as the specific physical laws $\Omega$ governing our dataset. For each transition, we curate specialized object pools containing entities that naturally exhibit the corresponding physical changes.

**Structured Generation.** Leveraging the transition categories, we synthesize videos utilizing a structured generation pipeline. We utilize GPT-5-mini to sample objects from the object pool and to instantiate a fixed *Wan Prompt* template used for video generation: [Start State] + [Trigger Event] + [Transition Description] + [Final State]. Using Wan2.2-T2V-A14B (Wan et al., 2025), we generate 1,000 raw videos per transition type. We include a static-camera constraint in the prompt so that pixel variations primarily reflect state $\{S_t\}$ transition rather than viewpoint changes. As shown in Figure 2(b), this pipeline successfully synthesizes diverse high-fidelity transitions across all defined physical domains.

**Camera Movement and Principle-driven Verification.** As illustrated in Figure 2(c), raw generations undergo a two-stage filtering process to ensure both viewpoint stability and physical correctness. First, we notice that current text-to-video models' instruction following ability is not perfect, leading to unwanted viewpoint shifts. We therefore apply ViPE (Huang et al., 2025) as a geometric stability filter. In practice, ViPE can confuse large non-rigid deformations with viewpoint changes. To reduce false rejections, we adopt an adaptive strategy that relaxes the ViPE threshold for transition types expected to induce substantial deformation. Next, we verify whether the video is consistent with its intended physical laws. Given the *Wan Prompt* and transition type, we prompt GPT-5-mini to propose $N$ (typically 3) transition-specific principles and classify them as *align*, *contradict*, or *unknown* based on the keyframes. We then calculate a verification score $S_{\text{verify}}$ and adopt a rigorous retention rule, preserving the video only if $S_{\text{verify}} = N_{\text{align}}/N_{\text{total}} \geq$

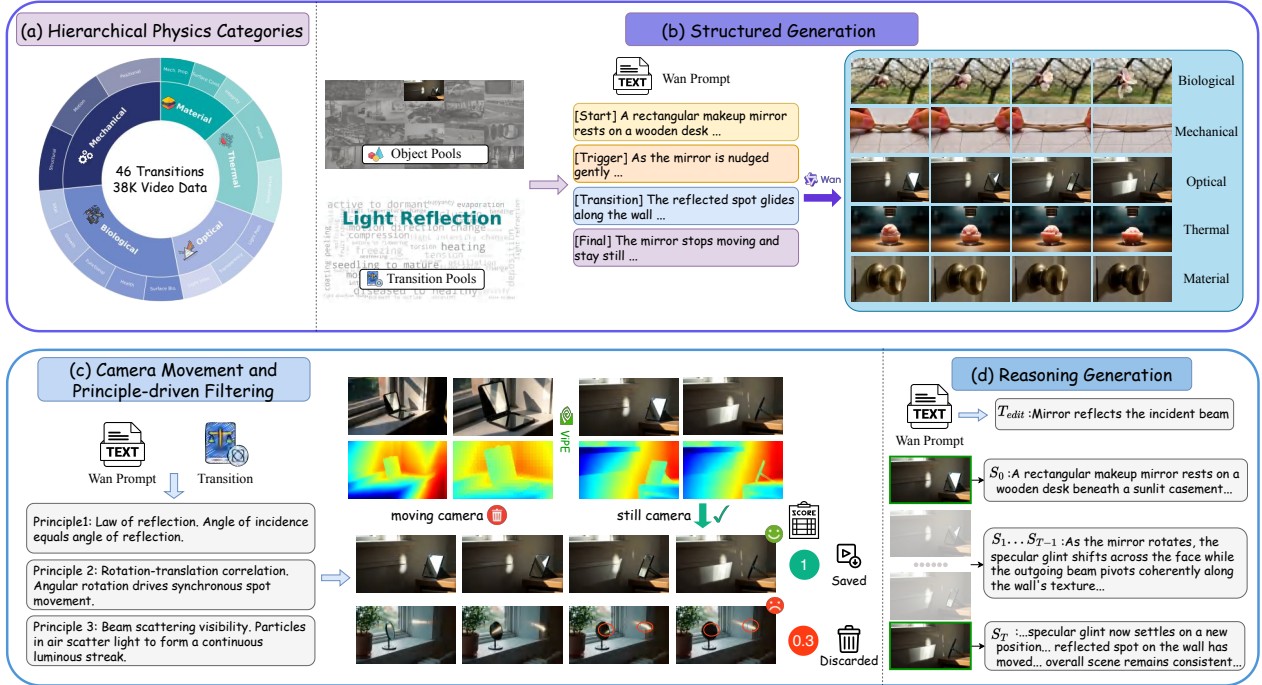

*Figure 2.* **Overview of the PhysicTran38K construction pipeline.** Starting from hierarchical physics categories, we synthesize videos using Wan2.2-T2V-A14B, filtered by ViPE with an adaptive strategy to preserve high-dynamic transitions. Candidate videos conduct principle-driven verification by GPT-5-mini, adhering to a rigorous retention rule. Finally, Qwen2.5-VL-7B performs constraint-aware annotation, generating instructions and structured reasoning while incorporating verification results to prevent hallucinations.

0.5. Crucially, rather than simply discarding flawed parts, we record contradicted principles as *negative evidence* to be used as constraints during the reasoning generation phase.

**Constraint-Aware Reasoning Generation.** Finally, we convert each retained video into training supervision that matches our formulation in Section 3.1. From each clip, we extract the first and last frames to form an editing pair $(I_{src}, I_{tgt})$ We also sample intermediate keyframes along the video trajectory to serve as the ground truth for our visual guidance mechanism in Section 3.3. We then use Qwen2.5VL-7B (Bai et al., 2025) to generate an editing instruction $T_{edit}$ describing the trigger event and brief desired evolution, along with a structured transition reasoning that summarizes observable evidence of the initial state $S_0$, middle transition process $S_1...S_{T-1}$, and the final state $S_T$. To ensure text-visual consistency, we integrate the verification outcomes as logical constraints during annotation: principles classified as *align* are encouraged as mechanisms consistent with $\Omega$, while *contradicted* or *unknown* principles are treated as hard negative constraints and must be explicitly excluded. This constraint-aware annotation mitigates over-optimistic physical descriptions when samples contain artifacts, allowing us to retain more usable videos and yielding approximately 38,000 samples.

### 3.3. PhysicEdit Framework

We present PhysicEdit, an end-to-end physics-aware editing framework built upon the Qwen-Image-Edit backbone. To bridge the gap between static image editing and dynamic physical laws, we instantiate a textual-visual dual-thinking mechanism with two complementary branches: physically-grounded reasoning, which uses a frozen Qwen2.5-VL-7B to provide explicit physical constraints as textual context, and implicit visual thinking, which introduces learnable transition queries to implicitly reconstruct transition priors in the latent space.

**Physically-Grounded Reasoning.** This branch produces a detailed physics reasoning that equips the following editing with explicit constraints before visual generation. Given the source image and the editing instruction, the frozen Qwen2.5-VL generates a structured physics reasoning trace that describes: (i) what physical laws and constraints must be respected, (ii) how the change should unfold causally in the scene, and (iii) how the relevant materials should behave throughout the transition. During training, we use the precomputed constraint-aware reasoning annotations from PhysicTran38K and ask Qwen2.5-VL to generate the reasoning trace on the fly during inference. This step serves as contextual grounding for the subsequent generation, while requiring no parameter updates to the MLLM.

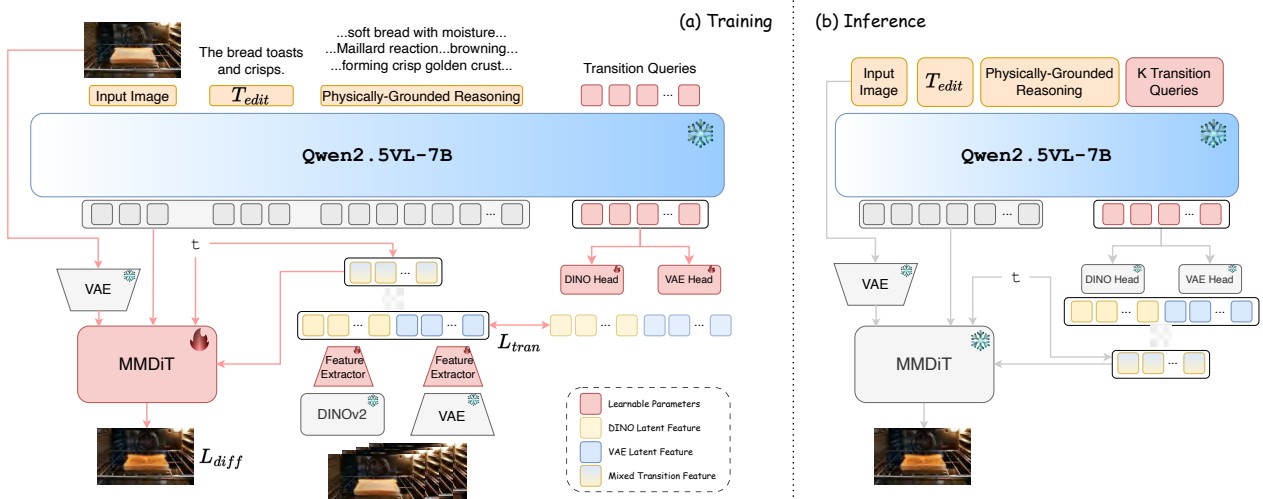

*Figure 3.* **Overview of the PhysicEdit framework. (a) Training:** We distill physical transition priors from video data into learnable transition queries. These queries are supervised by complementary visual features extracted from intermediate keyframes. **(b) Inference:** PhysicEdit follows a sequential workflow. The frozen MLLM first generates physically-grounded reasoning, which is then concatenated with the learned transition queries to serve as the condition for the diffusion backbone.

**Implicit Visual Thinking.** While text reasoning provides logical constraints, precise physical rendering requires visual transition priors that are not explicitly observable from the input image or text reasoning. To avoid modeling entire video frames which is cumbersome, we introduce *Implicit Visual Thinking*, as illustrated in Figure 3.

Inspired by MetaQuery (Pan et al., 2025), we append $K$ (typically 64) learnable transition queries immediately after the reasoning prefix. These queries are processed by the frozen Qwen2.5-VL together with the text and source image, producing context-aware query embeddings. During training, we supervise the query embeddings with *pseudo-target* transition features extracted from the associated video in PhysicTran38K. Specifically, we sample intermediate keyframes and encode them using two frozen visual encoders: DINOv2 for semantic structure and the Qwen-Image-Edit VAE for fine-grained texture. We then compress the encoder outputs into fixed-length pseudo-target embeddings, denoted as $F_{\text{DINO}}$ and $F_{\text{VAE}}$, using learnable feature extractors. In parallel, we map the transition-query embeddings through two projection heads to obtain the corresponding predictions, $\hat{F}_{\text{DINO}}$ and $\hat{F}_{\text{VAE}}$. We train the transition queries by aligning these predictions to the pseudo-targets. In this way, the transition queries learn to implicitly represent the missing evolution between $I_{src}$ and $I_{tgt}$, so that at inference time they can be instantiated solely from the source image, the instruction, and the physically-grounded reasoning. For more details, please refer to Appendix B.2.

**Timestep-Aware Dynamic Modulation.** Since diffusion models typically follow a coarse-to-fine trajectory, where global structure is formed before local texture is refined, we mix structure- and texture-level transition guidance ac-

cording to the diffusion timestep. Let $t \in [0, 1]$ denote the diffusion timestep, where larger $t$ corresponds to higher noise. During training, we construct the transition guidance from the video-derived pseudo-targets:

$$F_{\text{tran}}^{(t)} = t \cdot F_{\text{DINO}} + (1 - t) \cdot F_{\text{VAE}}. \qquad (3)$$

When $t \to 1$, the guidance emphasizes $F_{\text{DINO}}$ to provide structurally consistent cues; when $t \to 0$, it emphasizes $F_{\text{VAE}}$ to refine texture-level details. $F_{\text{tran}}^{(t)}$ is then sent to MMDiT as condition. At inference time, the video-derived pseudo-targets are unavailable, and we instead instantiate the guidance from the transition queries' predictions:

$$\hat{F}_{\text{tran}}^{(t)} = t \cdot \hat{F}_{\text{DINO}} + (1 - t) \cdot \hat{F}_{\text{VAE}}. \qquad (4)$$

The timestep-conditioned transition feature is injected into the diffusion model together with the physically-grounded reasoning condition to guide physics-aware generation.

**End-to-End Optimization.** We jointly optimize the learnable transition queries, feature extractors, dual-stream projection heads, and the diffusion backbone. The training objective is formulated as a composite loss function:

$$\mathcal{L}_{\text{total}} = \mathcal{L}_{\text{diff}} + \alpha \mathcal{L}_{\text{tran}}, \qquad (5)$$

where $\mathcal{L}_{\text{diff}}$ is the standard flow-matching loss applied to the diffusion backbone. The transition loss $\mathcal{L}_{\text{tran}}$ is designed to distill knowledge from the video frames into the transition queries in a timestep-aware manner:

$$\mathcal{L}_{\text{tran}} = t \left\| F_{\text{DINO}} - \hat{F}_{\text{DINO}} \right\|_2^2 + (1 - t) \left\| F_{\text{VAE}} - \hat{F}_{\text{VAE}} \right\|_2^2, \qquad (6)$$

*Table 1.* **Quantitative comparison on PICABench-Superficial** evaluated by GPT-5 for instruction-based editing models, where LP, LSE, RFL, RFR, DFM, CSL, GST, LST denote Light propagation, Light Source Effects, Reflection, Refraction, Deformation, Causality, Global State Transition, Local State Transition, respectively.

| Models | LP ↑ | LSE ↑ | RFL ↑ | RFR ↑ | DFM ↑ | CSL ↑ | GST ↑ | LST ↑ | Overall ↑ |
|---|---|---|---|---|---|---|---|---|---|
| *Proprietary Models* | | | | | | | | | |
| Nano Banana | 60.29 | 59.30 | 65.94 | 53.95 | 59.90 | 55.27 | 60.60 | 59.88 | 59.87 |
| GPT-Image-1 | 61.26 | 66.04 | 62.39 | 59.21 | 59.66 | 52.88 | 70.75 | 59.04 | 61.08 |
| Seedream 4.0 | 62.71 | 65.50 | 65.77 | 53.51 | 59.17 | 53.45 | 65.12 | 66.11 | 61.91 |
| Nano Banana Pro | 60.53 | 70.62 | **70.32** | 57.02 | 64.79 | **58.65** | 72.74 | 70.27 | 66.16 |
| GPT-Image-1.5 | **62.95** | **73.15** | 68.13 | **62.28** | **65.53** | 58.51 | **74.39** | **71.52** | **67.05** |
| *Open-Source Models* | | | | | | | | | |
| Bagel | 46.97 | 39.35 | 49.41 | 42.54 | 44.25 | 39.24 | 46.80 | 49.27 | 45.07 |
| Bagel-Think | 49.88 | 50.40 | 47.05 | 43.42 | 49.88 | 38.68 | 45.70 | 50.94 | 46.48 |
| OmniGen2 | 49.64 | 48.79 | 56.49 | 39.04 | 44.74 | 39.80 | 51.10 | 39.09 | 46.79 |
| Hidream-E1.1 | 49.15 | 48.25 | 49.07 | 46.49 | 44.50 | 40.51 | 56.40 | 40.33 | 47.90 |
| Step1X-Edit | 45.04 | 47.44 | 53.46 | 34.21 | 45.72 | 42.90 | 55.85 | 46.57 | 48.23 |
| Flux.1 Kontext | 54.96 | 57.41 | 57.50 | 36.40 | 51.83 | 38.12 | 48.79 | 47.61 | 48.93 |
| Uni-CoT | 54.56 | 59.82 | 49.01 | 55.12 | 49.12 | 48.79 | 60.98 | 51.70 | 53.56 |
| ChronoEdit | 59.92 | 67.48 | 57.94 | 56.18 | 56.12 | 51.60 | 64.14 | 55.69 | 58.67 |
| Qwen-Image-Edit | 62.95 | 61.19 | 62.90 | 55.26 | 48.66 | 48.95 | 67.33 | 54.89 | 61.26 |
| PhysicEdit | **64.88** | **76.16** | **67.72** | **62.22** | **60.76** | **59.23** | **67.67** | **60.52** | **64.86** |

where $F_{\text{DINO}}, F_{\text{VAE}}$ are pseudo-target embeddings and $\hat{F}_{\text{DINO}}, \hat{F}_{\text{VAE}}$ are the corresponding predictions from transition queries. We set $\alpha = 1$ in all experiments. Notably, the diffusion loss $\mathcal{L}_{\text{diff}}$ updates only the diffusion transformer and the feature extractors, while the transition queries and their projection heads are updated *exclusively* by the transition loss $\mathcal{L}_{\text{tran}}$. Therefore, this enforces a disentangled gradient update for learning how to compress and predict visual features independently.

## 4. Experiments

### 4.1. Experimental Setup

**Implementation Details.** We build PhysicEdit on top of the Qwen-Image-Edit backbone and fine-tune it on Physic-Tran38K using LoRA (Hu et al., 2022). Unless otherwise specified, we run LoRA fine-tuning for a single epoch with learning rate $5 \times 10^{-5}$, batch size 1 per GPU, and LoRA rank 128. All experiments are conducted on $4\times$ NVIDIA A100 GPUs, and training takes approximately 12 hours under this setup. For further details, refer to Appendix B.

**Evaluation.** We evaluate PhysicEdit on two benchmarks, PICABench (Pu et al., 2025) and KRISBench (Wu et al., 2025d), which probe complementary aspects of instruction-based image editing: physical realism and knowledge-grounded reasoning. PICABench focuses on whether edits produce physically realistic effects beyond instruction

following, covering optics, mechanics, and state transitions, which directly matches our goal of physics-consistent state evolution. KRISBench serves as a diagnostic benchmark that categorizes editing tasks into factual, conceptual, and procedural knowledge types; in particular, its conceptual-knowledge tasks and the temporal-perception subset within factual knowledge are closely related to our physical-transition setting. Accordingly, we expect improvements to concentrate on these KRISBench categories, while other categories may change minimally.

**Comparison Baselines.** We extensively evaluate our method against a diverse set of state-of-the-art proprietary and open-source image editing models. For proprietary models, we include Nano Banana, Nano Banana Pro (Google, 2025), GPT-Image-1 (OpenAI, 2025a), GPT-Image-1.5 (OpenAI, 2025b), Seedream 4.0 (ByteDance, 2025b), Gemini 2.0 Flash (Kampf & Brichtova, 2025), Doubao (ByteDance, 2025a), and Step 3o vision (stepfun, 2025). For open-source models, we compare against leading instruction-based methods, including Flux.1 Kontext (Labs et al., 2025). For recent advanced unified models, we include BAGEL (Deng et al., 2025) (and its reasoning variant BAGEL-Think), OmniGen2 (Wu et al., 2025b), Hidream-E1.1 (Cai et al., 2025), Step1X-Edit (Liu et al., 2025), Uni-CoT (Qin et al., 2025), and our backbone model Qwen-Image-Edit (Wu et al., 2025a). We also include two concurrent works: ChronoEdit (Wu et al., 2025c), which addresses

*Table 2.* **Quantitative comparisons on KRIS**. ‡ indicates applying EditThinker on Qwen-Image-Edit. 0.0* indicates that the model was not evaluated on multi-image editing.

| Models | Factual Knowledge | | | | Conceptual Knowledge | | | Procedural Knowledge | | | Overall Score |
| --- | --- | --- | --- | --- | --- | --- | --- | --- | --- | --- | --- |
| | Attribute Perception | Spatial Perception | Temporal Perception | Average Score | Social Science | Natural Science | Average Score | Logical Reasoning | Instruction Decompose | Average Score | |
| *Proprietary Models* | | | | | | | | | | | |
| Doubao | 70.92 | 59.17 | 40.58 | 63.30 | 65.50 | 61.19 | 62.23 | 47.75 | 60.58 | 54.17 | 60.70 |
| Step 3o vision | 69.67 | 61.08 | 63.25 | 66.70 | 66.88 | 60.88 | 62.32 | 49.06 | 54.92 | 51.99 | 61.43 |
| Gemini-2.0 | 66.33 | 63.33 | 63.92 | 65.26 | 68.19 | 56.94 | 59.65 | 54.13 | 71.67 | 62.90 | 62.41 |
| GPT-4o | **83.17** | **79.08** | **68.25** | **79.80** | **85.50** | **80.06** | **81.37** | **71.56** | **85.08** | **78.32** | **80.09** |
| *Open-Source Models* | | | | | | | | | | | |
| FLUX.1 Kontext [Dev] | 64.83 | 60.92 | 0.00* | 53.28 | 48.94 | 50.81 | 50.36 | 46.06 | 39.00 | 42.53 | 49.54 |
| OmniGen2 | 59.92 | 52.25 | 54.75 | 57.36 | 47.56 | 43.12 | 44.20 | 32.50 | 63.08 | 47.79 | 49.71 |
| BAGEL | 64.27 | 62.42 | 42.45 | 60.26 | 55.40 | 56.01 | 55.86 | 52.54 | 50.56 | 51.69 | 56.21 |
| BAGEL-Think | 67.42 | 68.33 | 58.67 | 66.18 | 63.55 | 61.40 | 61.92 | 48.12 | 50.22 | 49.02 | 60.18 |
| Uni-CoT | 72.76 | 72.87 | 67.10 | 71.85 | 70.81 | 66.00 | 67.16 | 53.43 | 73.93 | 63.68 | 68.00 |
| ChronoEdit | **79.22** | 73.08 | **81.22** | 78.18 | 74.20 | 71.49 | 72.14 | 57.97 | 61.17 | 59.35 | 70.96 |
| EditThinker ‡ | 78.48 | 73.83 | 0.00* | 77.24 | **76.20** | 70.69 | 72.02 | **65.23** | 66.89 | **65.94** | 71.91 |
| Qwen-Image-Edit | 75.48 | 80.58 | 71.73 | 76.00 | 65.78 | 59.75 | 61.22 | 52.25 | 71.76 | 60.61 | 65.56 |
| PhysicEdit | 77.64 | **81.67** | 76.13 | **78.29** | 74.49 | **71.57** | **72.27** | 59.02 | **70.81** | 64.06 | **72.16** |

*Table 3.* Ablation Study on physically-grounded reasoning and implicit visual thinking effectiveness.

| Model | Mechanics | Optics | State | Overall |
| --- | --- | --- | --- | --- |
| Qwen-Image-Edit | 55.04 | 64.29 | 62.85 | 61.26 |
| + SFT | 53.43 | 66.33 | 63.33 | 61.79 |
| + Physical Reasoning | 57.43 | 64.15 | 64.21 | 62.31 |
| + Visual Thinking | 52.38 | 68.05 | 64.05 | 62.41 |
| PhysicEdit | **59.79** | **68.19** | **65.10** | **64.86** |

*Table 4.* Ablation Study on DINO and VAE feature effectiveness.

| Model | Mechanics | Optics | GST | LST | Overall |
| --- | --- | --- | --- | --- | --- |
| Only DINO | 54.73 | 66.61 | **70.16** | 57.50 | 63.00 |
| Only VAE | 58.31 | 65.45 | 66.23 | 60.40 | 63.05 |
| Hard Switching | 57.39 | 65.43 | 68.84 | 62.08 | 63.52 |
| PhysicEdit | **59.79** | **68.19** | 67.67 | **60.52** | **64.86** |

image editing by leveraging video generation models, and EditThinker (Li et al., 2025a), which trains a specialized reasoning MLLM to critique generated results and refine instructions for the backbone model.

### 4.2. Main Results

**Performance on Physical Realism.** As shown in Table 1, PhysicEdit achieves an overall score of 64.86, establishing a new state-of-the-art among open-source models. Compared to the baseline Qwen-Image-Edit-2509, PhysicEdit improves all physical dimensions, supporting the effectiveness of our textual-visual dual-stream mechanism. Notably, the largest gains occur in categories requiring implicit dy-

namics: Light Source Effects increases from 61.19 to 76.16, Deformation rises by 12.0 points to 60.76, and Causality improves from 48.95 to 59.23. PhysicEdit also yields consistent improvements on Refraction and Local State Transition, indicating stronger adherence to global optical constraints and fine-grained state transition.

**Performance on Physics-Related Knowledge.** Table 2 presents the evaluation results on KRISBench. PhysicEdit achieves an overall score of 72.16, surpassing all open-source baselines and outperforming proprietary models such as Gemini-2.0 and Doubao. Consistent with our formulation of editing as a state transition, the improvements are clearly concentrated in categories that demand dynamic physical understanding. In the Factual Knowledge domain, the Temporal Perception score improves from 71.73 to 76.13, suggesting that our video-based training has endowed the model with a stronger sense of time evolution. Furthermore, within the Conceptual Knowledge domain, the Natural Science score improves by 11.9 points to 71.57. This validates that the physically-grounded reasoning stream effectively activates the model's latent scientific knowledge, enabling it to handle edits governed by strict natural laws rather than relying solely on surface-level visual patterns.

**Human Preference Study.** To complement automatic evaluation, we further conduct a human preference study on a subset of PICABench. We compare PhysicEdit with Qwen-Image-Edit, ChronoEdit, and Flux.1 Kontext across all six pairwise model combinations, using 100 samples per pair and 7 independent annotators per comparison. Annotators are shown the source image, the editing instruction,

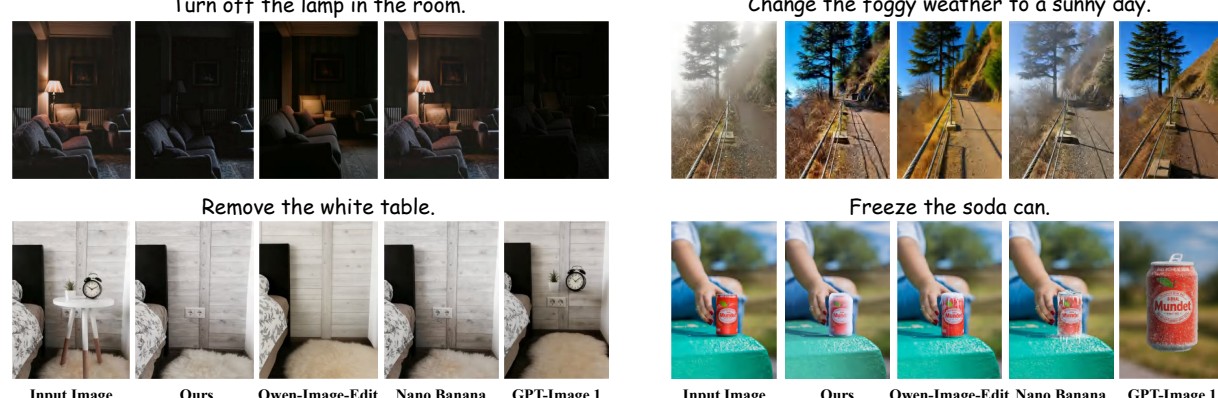

*Figure 4.* **Qualitative comparison on PICABench.** We visualize editing results across diverse physical domains, including Optics, Mechanics, Global State, and Local State. Compared to the backbone Qwen-Image-Edit and proprietary models, PhysicEdit consistently generates more physically plausible and visually natural results, avoiding the physical inconsistencies observed in baseline methods.

and two anonymized model outputs, and asked to select the edited image that better completes the instruction while considering both editing accuracy and physical plausibility. As summarized in Appendix E, PhysicEdit obtains the highest Elo rating and wins against all competing models in head-to-head comparisons, with an overall win rate of 60.0% across its 300 direct comparisons. These results confirm that the gains measured by automatic metrics are also reflected in human judgments of physical plausibility.

**Qualitative Analysis.** To demonstrate the physical fidelity of PhysicEdit, we visualize representative cases from PICABench in Figure 4. These examples span all the physical domains of PICABench, including Optics, Mechanics, and State Transitions. For example, in the "Turn off the lamp" case, our model correctly renders the global illumination decay and shadow propagation, whereas comparison models often struggle to maintain lighting consistency or merely darken the image globally without geometric correctness. These results support our quantitative findings, confirming that our physics-aware training enables the model to generate edits that are not only semantically aligned but also visually natural and physically law-grounded. Further detailed results are shown in Appendix D and Appendix G.

### 4.3. Ablation Studies

To validate our proposed components, we conduct comprehensive ablation studies on the PICABench dataset. Following PICABench's rule, we group LP, LSE, RFL, and RFR as Optics; DFM and CSL as Mechanics; GST and LST as State. The results are shown in Table 3 and Table 4. We provide more analysis and discussions with both quantitative and qualitative results in Appendix F.

**Effectiveness of Textual-Visual Dual-Stream Thinking.** We first investigate the necessity of our architectural design by comparing it against a standard supervised fine-tuning baseline. As shown in Table 3, simply fine-tuning Qwen-Image-Edit on our dataset yields only a marginal gain and exposes a trade-off. Optics increases, whereas Mechanics deteriorates, indicating that data-only supervision is insufficient to reliably induce abstract physical constraints. Incorporating Physically-Grounded Reasoning significantly boosts the Mechanics score to 57.43, confirming that explicit textual planning is essential for enforcing logical physical constraints. However, Optics changes only marginally, suggesting limited effectiveness for optics-dominant effects when relying on textual constraints alone. In contrast, using only Implicit Visual Thinking markedly strengthens Optics (68.05), implying that learnable transition queries provide effective visual guidance; yet Mechanics drops to 52.38 in the absence of textual constraints, reflecting weakened logical consistency. The full PhysicEdit achieves the highest performance across all metrics, demonstrating that the two streams are not redundant but deeply complementary: the textual stream ensures logical plausibility, while the visual stream handles the rendering of dynamic state transitions.

**Effectiveness of Implicit Visual Thinking Design.** We further analyze the architectural design of our implicit visual thinking module, with emphasis on whether both DINOv2 and VAE features are necessary. Table 4 compares our dual-stream approach against single-stream variants, where we split State into GST and LST for more detailed analysis. Using DINO features alone yields the highest GST of 70.16, confirming its structural strength, yet it suffers in Mechanics due to insensitivity to local deformations. Conversely, VAE features improve texture-related LST to 60.40 but lack global coherence. To notice, our model performs slightly

worse on GST, which reflects a necessary trade-off: we relax DINO's excessive structural rigidity to enable substantial physical deformations, thereby achieving superior overall physical fidelity.

We then study the effectiveness of our fusion strategy by comparing it with a Hard Switching baseline, which rigidly utilizes DINO for the structural formation stage ($t \in [1, 0.7]$) and abruptly switches to VAE for texture refinement ($t < 0.7$). While this strategy outperforms single-stream variants, it still falls short of our full model. We attribute this to the continuous nature of the diffusion process, where a hard cutoff introduces a semantic discontinuity in the guidance signal. In contrast, PhysicEdit employs Timestep-Aware Modulation to smoothly interpolate between structural and textural guidance. This alignment with the inherent coarse-to-fine trajectory of diffusion enables the model to attain the best overall performance of 64.86.

## 5. Conclusion

We presented a paradigm shift in image editing by modeling the process as a continuous Physical State Transition. We introduce PhysicTran38K, a video-based dataset curated with principle-driven verification and constraint-aware annotation, and propose PhysicEdit, which combines physically grounded reasoning with latent transition queries to leverage video supervision while preserving standard single-image inference. Our method significantly outperforms existing baselines in physical plausibility, validating the effectiveness of transition-centric supervision. This work underscores the importance of physical laws in visual generation and provides a robust framework for future physics-aware generation research.

## Impact Statement

This paper presents work whose goal is to advance the field of Machine Learning, specifically in enhancing the physical plausibility of generative editing. Our framework, PhysicEdit, and the constructed dataset, PhysicTran38K, significantly improve the visual realism of edited content by enforcing adherence to physical laws. While this advancement offers substantial benefits for creative industries, virtual prototyping, and education, we acknowledge that the increased realism of manipulated images could potentially be misused to create misleading content or misinformation that is harder to distinguish from reality. We advocate for the responsible use of this technology and encourage future research into detection methods that can identify synthetic physical inconsistencies. There are no other specific societal consequences that we feel must be highlighted here.

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

## A. Detailed Dataset Statistics and Taxonomy

In this section, we provide a granular breakdown of the PhysicTran38K taxonomy and the distribution of the collected video data.

### A.1. Statistical Overview

As summarized in Table 5, our dataset is constructed based on a three-level hierarchy: 5 primary physical domains, 16 intermediate sub-domains, and 46 distinct Transition Types. After our rigorous filtering pipeline, we retained a total of 38,620 high-quality video-instruction pairs. The distribution across domains is relatively balanced, ensuring that the model learns a diverse set of physical laws ranging from rigid body mechanics to biological growth.

*Table 5.* Detailed statistics of PhysicTran38K across five primary physical domains. The video counts represent the final data used for training after filtering.

| Primary Domain | Sub-domains | Transition Types | Filtered Videos |
|---|---|---|---|
| **Mechanical** | 3 | 12 | 10,245 |
| **Biological** | 5 | 12 | 10,242 |
| **Thermal** | 2 | 8 | 6,602 |
| **Optical** | 3 | 8 | 6,245 |
| **Material** | 3 | 6 | 5,286 |
| **Total** | **16** | **46** | **38,620** |

### A.2. Hierarchical Taxonomy Definition

To ensure the coverage of fundamental physical interactions, we define specific intermediate sub-domains for each primary category. The complete taxonomy is organized as follows:

**1. Mechanical State.** This domain covers the kinematics and dynamics of rigid and non-rigid bodies. It is further divided into 3 sub-domains:

- **Positional:** Changes in spatial location or orientation driven by external forces. *Transitions: Translation, Rotation, Oscillation, Gravity, Buoyancy.*

- **Motion:** Changes in velocity or momentum patterns. *Transitions: Motion Direction Change.*

- **Structural:** Physical alterations to the object's structure. *Transitions: Compression, Tension, Bending, Torsion, Fracture, Collapse.*

**2. Biological State.** This domain focuses on the life processes of organic entities. It comprises 5 sub-domains:

- **Vital:** Fundamental signs of life or death. *Transitions: Alive to Dead.*

- **Growth:** Expansion or development over time. *Transitions: Decay, Fruit Ripening, Germination, Mature to Flower, Seedling to Mature.*

- **Functional:** Active behaviors or movements specific to living organisms. *Transitions: Active to Dormant, Dormant to Active.*

- **Health:** Indications of well-being or sickness. *Transitions: Disease to Health, Health to Disease.*

- **Surface Biological:** External biological changes. *Transitions: Mold Growth, Moss Algae Growth.*

**3. Optical State.** This domain captures how light interacts with matter. It includes 3 sub-domains:

- **Light Interaction:** Modifications to the radiometric, spectral, and spatial properties of light sources. *Transitions: Light Color Change, Light Direction Change, Light Intensity Change, Light Temperature Change.*

- **Transparency:** Changes in opacity or translucency. *Transitions: Clear to Translucent.*

- **Light Path Change:** Geometric optics phenomena. *Transitions: Light Reflection, Light Refraction.*

**4. Thermal State.** This domain governs thermodynamic processes and heat transfer. It consists of 2 sub-domains:

- **Temperature:** Visible effects of heating or cooling. *Transitions: Cooling, Heating.*

- **Phase:** Transitions between solid, liquid, and gas states. *Transitions: Condensation, Deposition, Evaporation, Freezing, Melting, Sublimation.*

**5. Material State.** This domain describes the intrinsic physical properties of matter. It is categorized into 3 sub-domains:

- **Integrity:** The structural wholeness and physical continuity of the material volume. *Transitions: Intact to Broken.*

- **Surface Condition:** Degradation or physical alteration of the material's exterior layer due to wear or exposure. *Transitions: Coating Peeling, Abrasion, Weathering.*

- **Mechanical Property:** Variations in the material's rigidity, compliance, and resistance to deformation. *Transitions: Hardening, Softening.*

We provide some detailed data illustrations in Figure 5 and Figure 6.

## B. Model Setup Details

### B.1. Model Specifications and Assets

To ensure the reproducibility of our experiments and facilitate future research, we detail the specific versions and source URLs of all foundation models and codebases employed in our framework.

**Codebase and Backbone.** Our implementation is developed based on **DiffSynth Studio**[1], a high-efficiency diffusion framework. Specifically, our PhysicEdit is built upon the **Qwen-Image-Edit-2509**[2]. We use this version as the initialization for our diffusion backbone and keep it frozen during the visual encoder training phase to preserve its original text-following capabilities.

**Data Construction Model.** For the data construction pipeline, specifically in the generation of prompts, transition principles, and verification steps, we utilize the **GPT-5-mini**[3]. This model serves as the core reasoning engine for ensuring the physical correctness of our collected transitions.

**Visual Encoders.** Our implicit visual thinking module leverages two distinct encoders to capture complementary information. For *structural semantics*, we employ the **DINOv2-Base with Registers**[4], which provides robust geometric and layout priors. For *fine-grained texture*, we utilize the specific **VAE** component[5] from the Qwen-Image-Edit repository to ensure precise alignment with the backbone's latent space.

---

[1] https://github.com/modelscope/DiffSynth-Studio
[2] https://huggingface.co/Qwen/Qwen-Image-Edit-2509
[3] https://platform.openai.com/docs/models/gpt-5-mini
[4] https://huggingface.co/facebook/dinov2-with-registers-base
[5] https://huggingface.co/Qwen/Qwen-Image-Edit-2509/tree/main/vae

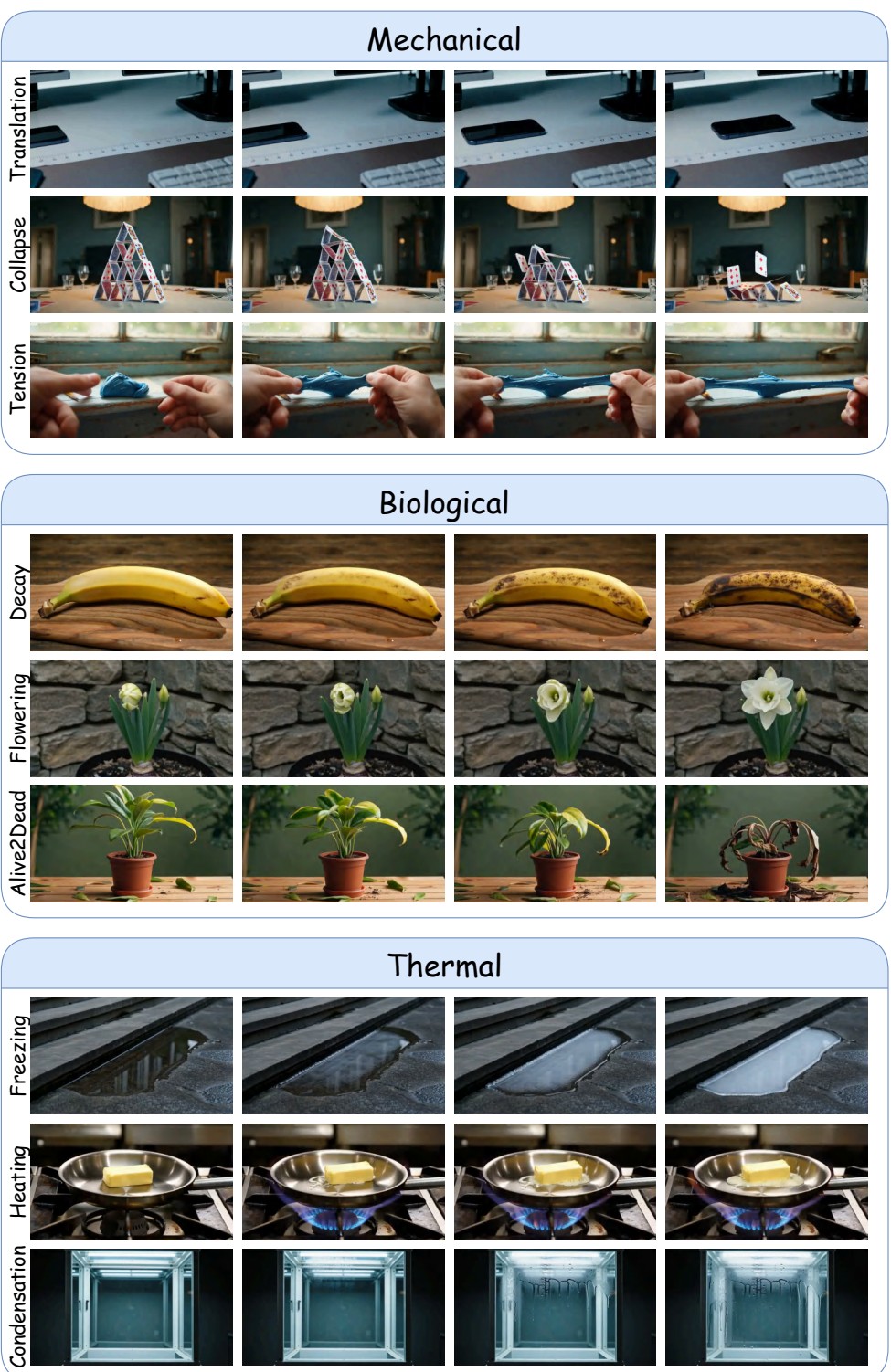

*Figure 5.* More detailed illustrations on Mechanical, Biological and Thermal data.

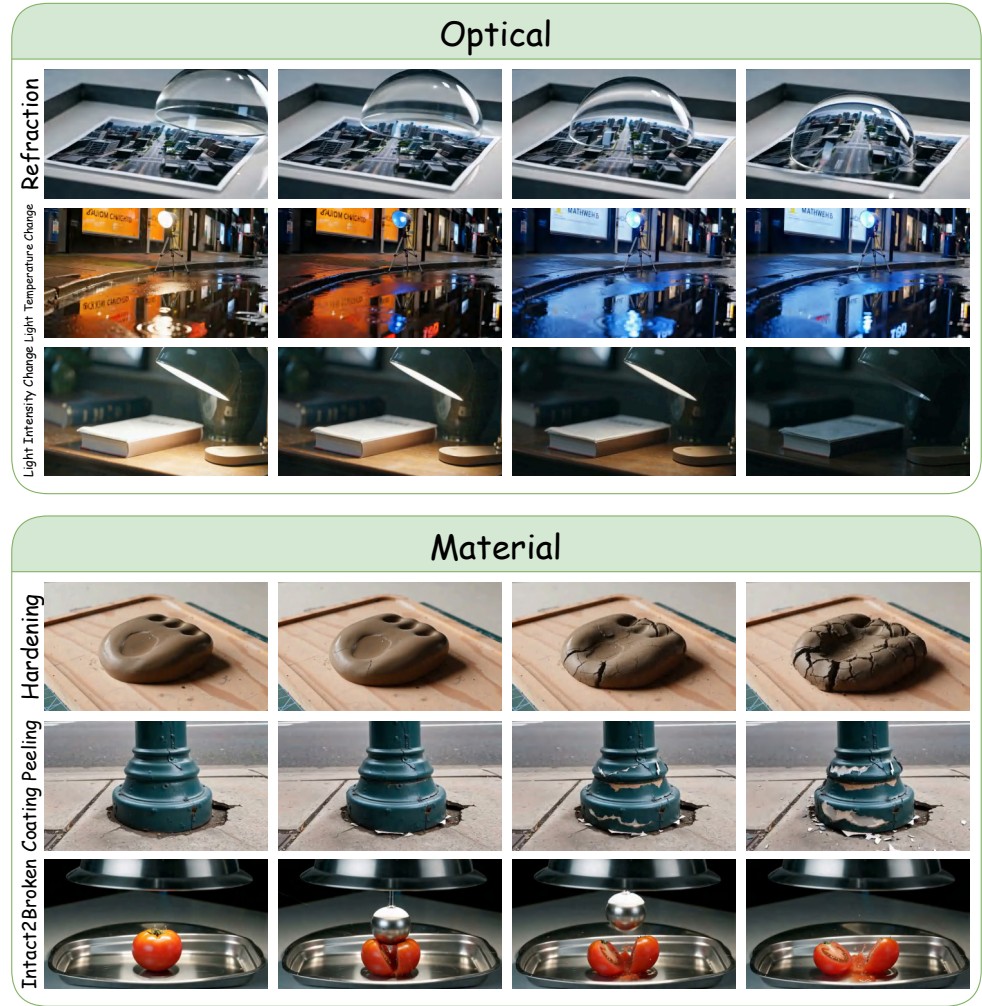

*Figure 6.* More detailed illustrations on Optical and Material data.

## B.2. Implementation Details of Implicit Visual Thinking

In this section, we provide further details regarding the architectural implementation and training strategies of the Implicit Visual Thinking module.

**Transition Query Injection and Parameterization.** The $K$ transition queries are implemented as a sequence of learnable special tokens. During the forward pass, these tokens are concatenated to the input sequence immediately following the edit instruction and the generated physically-grounded reasoning text. The combined sequence is then processed by the frozen Qwen2.5-VL, allowing the transition queries to attend to both the visual context of the source image and the textual reasoning cues. Crucially, these queries serve as global learnable parameters shared across all data samples. While we explored the possibility of using domain-specific query banks (e.g., maintaining separate sets of queries for Optical vs. Mechanical transitions), empirical results indicated that such segregation led to overfitting on the training distribution and reduced generalization on external benchmarks. We believe that a shared global parameter set forces the model to learn more abstract and universal physical representations, thereby enhancing its robustness to diverse and unseen physical scenarios.

**Video Supervision Strategy.** To distill physical priors from the video data, we employ a uniform sampling strategy to select $N = 6$ intermediate keyframes from each video clip. Since our PhysicTran38K is constructed via a rigorous principle-driven generation pipeline, the physical state transitions are distributed consistently throughout the video duration.

Consequently, the physical information contained in the video is rich and temporally balanced. Our preliminary experiments suggested that simple uniform sampling is sufficient to capture these dynamics, and more complex sampling strategies (e.g., motion-based or attention-based selection) yielded negligible performance gains in our setting.

**Feature Extraction and Transition Delta.** To provide explicit supervision for the learnable transition queries, our feature extractors are designed to capture dynamic evolution rather than static image content. Specifically, given N intermediate video frames, we first extract their raw visual representations using the frozen DINOv2 and VAE encoders. To align these high-dimensional spatial features with our 1D transition queries, we employ a Perceiver Resampler-like mechanism, which compresses the encoder outputs into a fixed-length sequence of $K$ tokens. Crucially, since our objective is to learn the *change* in physical states, we process the source image through this exact same pipeline to obtain a baseline latent representation. The final supervision signal is then formulated as the residual difference between the compressed features of the intermediate frame and those of the source image. By explicitly targeting this latent delta, we force the transition queries to focus exclusively on the physical dynamics and transformations, effectively bypassing the redundant reconstruction of static background information.

## C. Detailed Related Works

### C.1. Instruction-Based Image Editing

The field of image editing has witnessed a paradigm shift from domain-specific Generative Adversarial Networks (Goodfellow et al., 2020) to high-fidelity Diffusion Models (Ho et al., 2020; Song et al., 2021; BAKR et al.). Early diffusion-based approaches (Hertz et al., 2022; Zhao et al., 2023) primarily manipulate cross-attention maps or invert latents to balance content preservation with editing strength, yet they often lacked precise controllability for complex structural changes. This limitation catalyzed the emergence of instruction-tuned methods (Brooks et al., 2023; Labs, 2024; Wei et al., 2024; Zhuo et al., 2025b) which explicitly learn the mapping between input images and textual instructions. Recent advancements have culminated in unified multimodal frameworks that integrate vision and language modalities for robust instruction alignment. A representative model is Qwen-Image-Edit (Wu et al., 2025a), which features a dual-stream architecture composing a frozen Qwen2.5-VL (Bai et al., 2025) and a Multi-Modal Diffusion Transformer (MMDiT) (Esser et al., 2024). In this design, the MLLM-encoded representations replace standard text embeddings to serve as the condition input for the MMDiT. Despite their remarkable success in semantic manipulation, current paradigms prioritize perceptual quality while largely neglecting fundamental physical laws.

### C.2. Video Prior for Image Editing

Recent research has increasingly explored leveraging video data to enhance image editing. While early attempts (Deng et al., 2025; Xiao et al., 2024; Chen et al., 2025) primarily constructed training pairs from video keyframes to enhance consistency, recent works (Wu et al., 2025c; Rotstein et al., 2025) have shifted towards a generative reasoning paradigm. Notably, the concurrent work ChronoEdit (Wu et al., 2025c) adapts video generation models to *explicitly* synthesize intermediate frames as reasoning steps for editing. However, such explicit pixel-level generation is computationally demanding and prone to error accumulation. Distinct from these approaches, we propose an *implicit* paradigm: instead of generating full video frames, we distill physical state transition laws into compact latent queries. This allows our model to simulate dynamics flexibly and efficiently in the feature space, bypassing the heavy burden of explicit video synthesis while retaining the physical fidelity derived from temporal data.

## D. General Editing and Reasoning Editing Results

**Performance on General Image Editing.** To ensure that acquiring physical priors does not compromise the fundamental semantic capabilities of the backbone, we evaluate PhysicEdit on standard general image editing benchmarks, namely ImgEdit-Bench and GEdit-Bench-EN. As detailed in Table 6, PhysicEdit not only maintains but consistently improves upon the performance of the base Qwen-Image-Edit model. Specifically, the overall score on ImgEdit-Bench increases from 4.35 to 4.40, and the overall score (G_O) on GEdit-Bench-EN rises from 7.56 to 7.87. It demonstrates robust performance across diverse fundamental operations, including addition, replacement, and style transfer, remaining highly competitive among state-of-the-art open-source models. This confirms that our transition-centric fine-tuning strategy effectively injects dynamic physical knowledge without causing catastrophic forgetting of the backbone's original text-alignment and semantic editing proficiency.

*Table 6.* Comparison results on general image editing, including ImgEdit-Bench and GEdit-Bench-EN. ‡ indicates applying EditThinker on Qwen-Image-Edit. All use GPT-4.1 for evaluation.

| Model | ImgEdit-Bench | | | | | | | | | | GEdit-Bench-EN | | |
|---|---|---|---|---|---|---|---|---|---|---|---|---|---|
| | Add | Adjust | Extract | Replace | Remove | Background | Style | Hybrid | Action | Overall | G_SC | G_PQ | G_O |
| *Proprietary Models* | | | | | | | | | | | | | |
| GPT-4o (OpenAI, 2025a) | 4.61 | 4.33 | 2.9 | 4.35 | 3.66 | 4.57 | 4.93 | 3.96 | 4.89 | 4.20 | 7.74 | 8.13 | 7.49 |
| *Open-source Models* | | | | | | | | | | | | | |
| OmniGen (Xiao et al., 2025) | 3.47 | 3.04 | 1.71 | 2.94 | 2.43 | 3.21 | 4.19 | 2.24 | 3.38 | 2.96 | 5.96 | 5.89 | 5.06 |
| Step1X-Edit (Liu et al., 2025) | 3.88 | 3.14 | 1.76 | 3.40 | 2.41 | 3.16 | 4.63 | 2.64 | 2.52 | 3.06 | 7.66 | 7.35 | 6.97 |
| BAGEL (Deng et al., 2025) | 3.56 | 3.31 | 1.70 | 3.30 | 2.62 | 3.24 | 4.49 | 2.38 | 4.17 | 3.20 | 7.36 | 6.83 | 6.52 |
| OmniGen2 (Wu et al., 2025b) | 3.57 | 3.06 | 1.77 | 3.74 | 3.20 | 3.57 | 4.81 | 2.52 | 4.68 | 3.44 | 7.16 | 6.77 | 6.41 |
| FLUX.1 Kontext [Dev] (Labs et al., 2025) | 3.76 | 3.45 | 2.15 | 3.98 | 2.94 | 3.78 | 4.38 | 2.96 | 4.26 | 3.52 | 6.52 | 7.38 | 6.00 |
| EditThinker ‡ (Li et al., 2025a) | 4.23 | 4.43 | 4.24 | 4.20 | 4.21 | 4.44 | 4.76 | 3.91 | 4.68 | 4.40 | 8.30 | 7.86 | 7.73 |
| UniWorld-V2 (Li et al., 2025b) | 4.29 | 4.44 | 4.32 | 4.69 | 4.72 | 4.41 | 4.91 | 3.83 | 4.83 | 4.49 | 8.39 | 8.02 | 7.83 |
| Qwen-Image-Edit | 4.32 | 4.36 | 4.04 | 4.64 | 4.52 | 4.37 | 4.84 | 3.39 | 4.71 | 4.35 | 8.00 | 7.86 | 7.56 |
| PhysicEdit | 4.36 | 4.32 | 3.95 | 4.71 | 4.57 | 4.35 | 4.88 | 3.54 | 4.91 | 4.40 | 8.68 | 7.54 | 7.87 |

*Table 7.* Comparison of model performance on RISE-Bench. ‡ indicates applying EditThinker on Qwen-Image-Edit.

| Model | Temporal | Causal | Spatial | Logical | Overall |
|---|---|---|---|---|---|
| *Proprietary Models* | | | | | |
| Seedream-4.0 | 12.9 | 12.2 | 11.0 | 7.1 | 10.8 |
| GPT-Image-1 | 34.1 | 32.2 | 37.0 | 10.6 | 28.9 |
| Gemini-2.5-Flash-Image | 25.9 | 47.8 | 37.0 | 18.8 | 32.8 |
| *Open-source Models* | | | | | |
| Step1X-Edit | 0.0 | 2.2 | 2.0 | 3.5 | 1.9 |
| Ovis-U1 | 1.2 | 3.3 | 4.0 | 2.4 | 2.8 |
| FLUX.1-Kontext-Dev | 2.3 | 5.5 | 13.0 | 1.2 | 5.8 |
| BAGEL | 2.4 | 5.6 | 14.0 | 1.2 | 6.1 |
| Qwen-Image-Edit | 4.7 | 10.0 | 17.0 | 2.4 | 8.9 |
| BAGEL-Think | 5.9 | 17.8 | 21.0 | 1.2 | 11.9 |
| EditThinker ‡ | 10.8 | 23.3 | 27.0 | 8.2 | 17.8 |
| Qwen-Image-Edit | 4.7 | 10.0 | 17.0 | 2.4 | 8.9 |
| PhysicEdit | 21.2 | 23.3 | 25.0 | 3.5 | 18.6 |

**Performance on Reasoning-Based Editing.** We further assess our model on RISE-Bench, a challenging benchmark designed to evaluate complex reasoning capabilities in editing tasks. As shown in Table 7, PhysicEdit achieves a substantial breakthrough, more than doubling the overall score of the base Qwen-Image-Edit from 8.9 to 18.6. Consistent with our physical state transition formulation, the most dramatic improvements are observed in dimensions that strictly demand dynamic and logical understanding: the Temporal score improves from 4.7 to 21.2, and the Causal score improves from 10.0 to 23.3. Notably, PhysicEdit attains the highest overall performance among all evaluated open-source models. These results validate that our textual-visual dual-thinking mechanism successfully equips the model with deep causal and temporal reasoning abilities, significantly narrowing the gap with leading proprietary models in complex physical scenarios.

# E. Human Evaluation Details

To further assess physical plausibility from a human perspective, we conduct a large-scale human preference study on a subset of PICABench using the Rapidata platform. We compare four models: PhysicEdit, Qwen-Image-Edit, ChronoEdit, and Flux.1 Kontext. We evaluate all six pairwise model combinations, with 100 samples per pair selected from PICABench. Each comparison is rated by 7 independent annotators, who are shown the source image, the editing instruction, and two anonymized model outputs side by side. Annotators are asked: *"Which edited image better completes the editing instruction? Consider both the accuracy of the edit and physical plausibility."* The winner of each comparison is determined by majority vote. In total, we collect 4,200 individual judgments across 600 pairwise comparisons. We then compute Elo ratings following the standard Elo rating protocol (Elo & Sloan, 1978), using $K = 24$, $\sigma = 400$, Gaussian-initialized ratings

with $\mu = 1000$ and $\sigma_0 = 300$, and $T = 100$ shuffled rounds for robustness.

*Table 8.* Elo ratings from human evaluation on PICABench.

| Rank | Model | Elo Rating |
|---|---|---|
| 1 | PhysicEdit | **1045** |
| 2 | Qwen-Image-Edit | 1022 |
| 3 | ChronoEdit | 989 |
| 4 | Flux.1 Kontext | 915 |

*Table 9.* Pairwise win rate matrix from human evaluation on PICABench. Each entry denotes the row model's win rate against the column model.

| | **PhysicEdit** | **Qwen-Image-Edit** | **ChronoEdit** | **Flux.1 Kontext** |
|---|---|---|---|---|
| PhysicEdit | – | **52.0%** | **56.0%** | **72.0%** |
| Qwen-Image-Edit | 48.0% | – | **57.0%** | **62.0%** |
| ChronoEdit | 44.0% | 43.0% | – | **64.0%** |
| Flux.1 Kontext | 28.0% | 38.0% | 36.0% | – |

As shown in Table 8 and Table 9, PhysicEdit achieves the highest Elo rating among the four models and is the only method that wins in head-to-head comparisons against all other models. Specifically, PhysicEdit achieves win rates of 72.0% against Flux.1 Kontext, 56.0% against ChronoEdit, and 52.0% against Qwen-Image-Edit. Across all 300 direct head-to-head comparisons involving PhysicEdit, its overall win rate is 60.0%. Overall, these human evaluation results align with the trends observed in automatic metrics and provide additional evidence that PhysicEdit's improvements in physical realism are perceptible to human annotators.

## F. Analysis and Discussion

*Table 10.* Comparison between explicit and implicit visual thinking.

| Model | PICA | KRIS |
|---|---|---|
| ChronoEdit | 58.67 | 70.96 |
| +SFT PhysicTran38K | 61.43 | 69.59 |
| PhysicEdit | **64.86** | **72.16** |

*Table 11.* Comparison between different data construction logic.

| Model | PICA | KRIS |
|---|---|---|
| Qwen-Image-Edit | 61.26 | 65.56 |
| +SFT PICA-100K | 62.96 | 66.00 |
| PhysicEdit | **64.86** | **72.16** |

**Explicit vs. Implicit Visual Thinking.** We further compare PhysicEdit with ChronoEdit (Wu et al., 2025c), a concurrent approach that follows an explicit visual thinking paradigm by synthesizing intermediate frames to represent transition trajectories. Such frame rollout is computationally intensive and can accumulate errors over long horizons (Figure 7). In contrast, PhysicEdit adopts implicit visual thinking by encoding transition dynamics into latent transition queries, avoiding pixel-level intermediate-frame synthesis at inference.

As shown in Table 10, PhysicEdit is better than ChronoEdit on both benchmarks, increasing PICABench from 58.67 to 64.86 and KRISBench from 70.96 to 72.16. To disentangle the effect of supervision from modeling choice, we fine-tune ChronoEdit on PhysicTran38K. This raises its PICABench score to 61.43 but reduces its KRISBench score to 69.59, suggesting a trade-off when adapting an explicit trajectory-based pipeline with transition-centric fine-tuning. PhysicEdit mitigates this trade-off by design: transition-specific learning is localized in lightweight transition queries and adapters while keeping the VLM frozen, which helps incorporate physical priors without compromising the backbone's broad generalization capabilities.

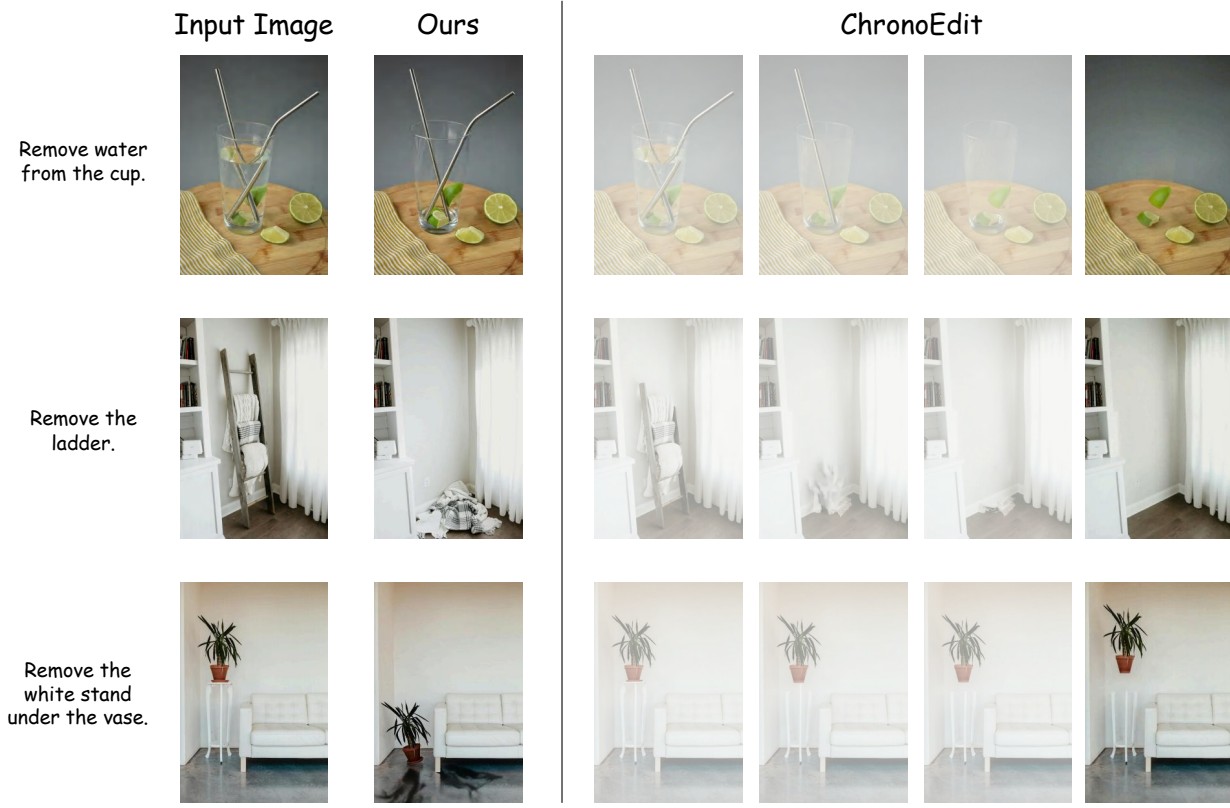

*Figure 7.* **Comparison with explicit visual thinking (ChronoEdit).** ChronoEdit employs an explicit paradigm that synthesizes intermediate video frames to model the editing process. As visualized in the rollouts, this pixel-level generation suffers from severe **error accumulation** over long horizons, causing the final results to degenerate (e.g., background distortion or object disappearance). In contrast, by encoding dynamics into latent transition queries (*Implicit Visual Thinking*), our method avoids these synthesis errors, maintaining structural integrity and yielding precise, high-fidelity edits.

**Principle Acquisition vs. Instruction Overfitting.** A critical question is whether the improvements stem from learning fundamental physical laws or merely overfitting to the distribution of editing instructions. To investigate this, we compare our data construction strategy against PICA-100K, a dataset specifically curated for the PICABench benchmark. As shown in Table 11, when fine-tuning the base Qwen-Image-Edit on PICA-100K, the improvements are **marginal**: the PICABench score rises slightly from 61.26 to 62.96, and the KRISBench score sees a negligible increase from 65.56 to 66.00. This suggests that the PICA-100K construction primarily teaches the model to mimic surface-level editing patterns rather than internalizing underlying dynamics. In contrast, training on PhysicTran38K with our PhysicEdit framework boosts performance substantially to **64.86** on PICABench and **72.16** on KRISBench. This confirms that our hierarchical, principle-driven data construction enables true *Principle Acquisition*. By exposing the model to pure state transitions (e.g., optical paths, material deformations) rather than just editing pairs, the model internalizes the "physics of change," allowing it to generalize to the diverse and complex scenarios presented in knowledge-grounded benchmarks like KRIS.

## G. More Qualitative Results

Figures 8, 9, and 10 present more detailed comparison results on PICABench.

**Superficial Prompt:** Remove the yellow chair

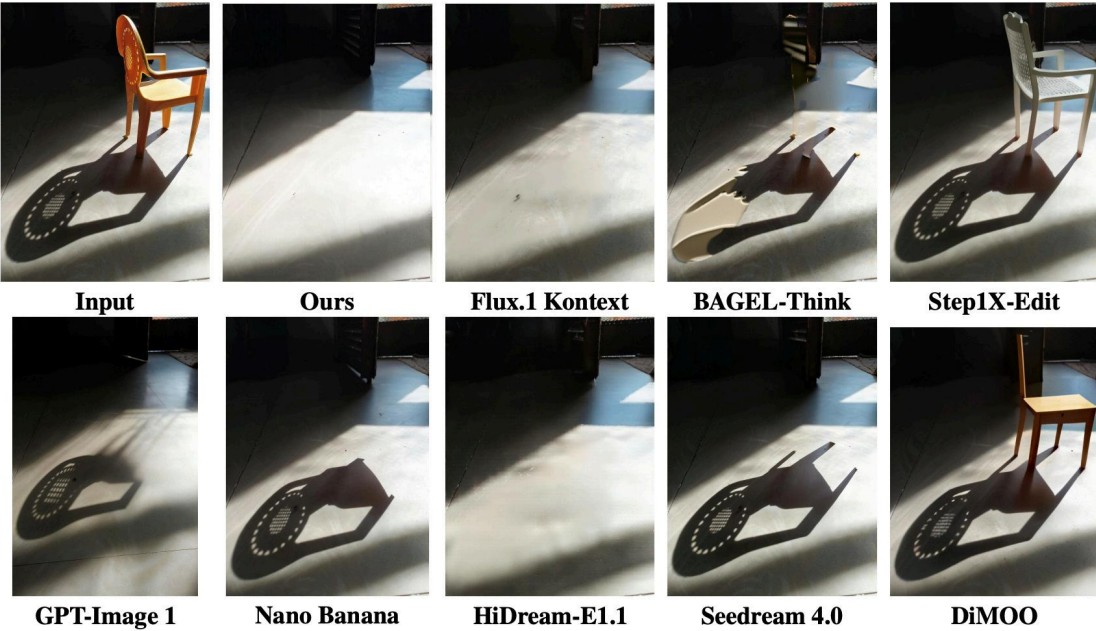

**Superficial Prompt:** Move the potted plant to left side of the table.

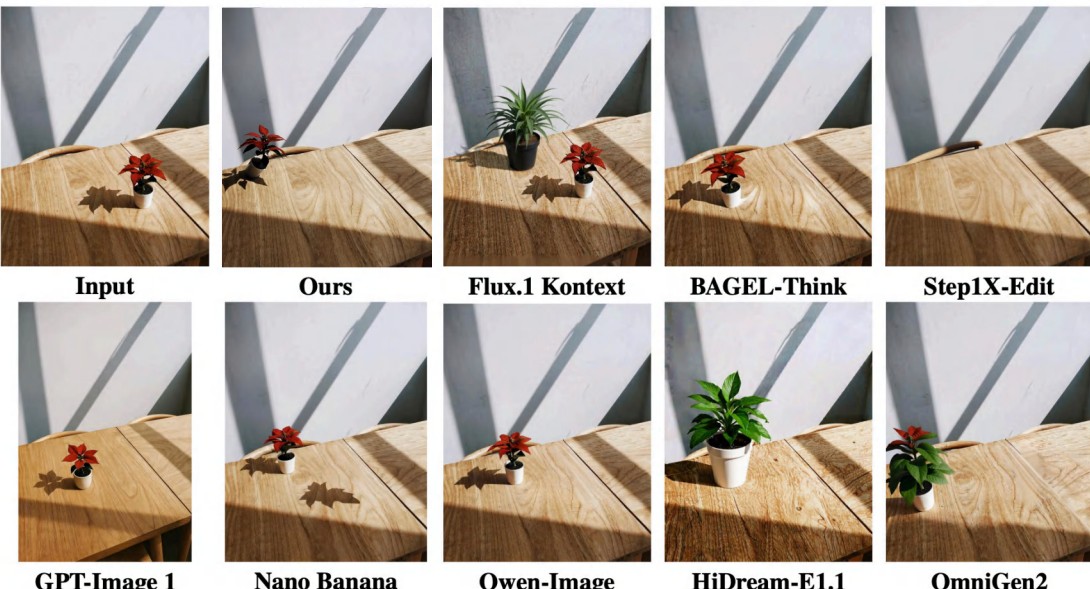

*Figure 8.* More qualitative results on PICABench, showing light propogation related results.

**Superficial Prompt:** Turn on the lamp on the bedside table.

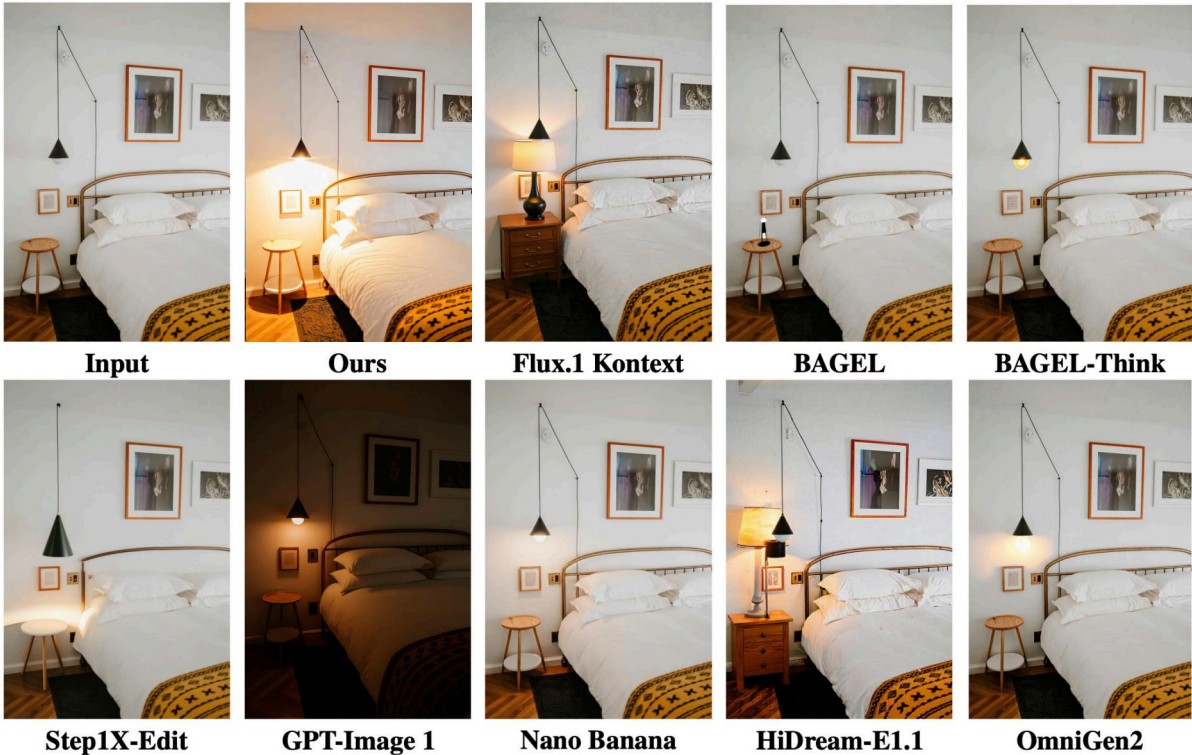

**Superficial Prompt:** Turn off the lamp in the room.

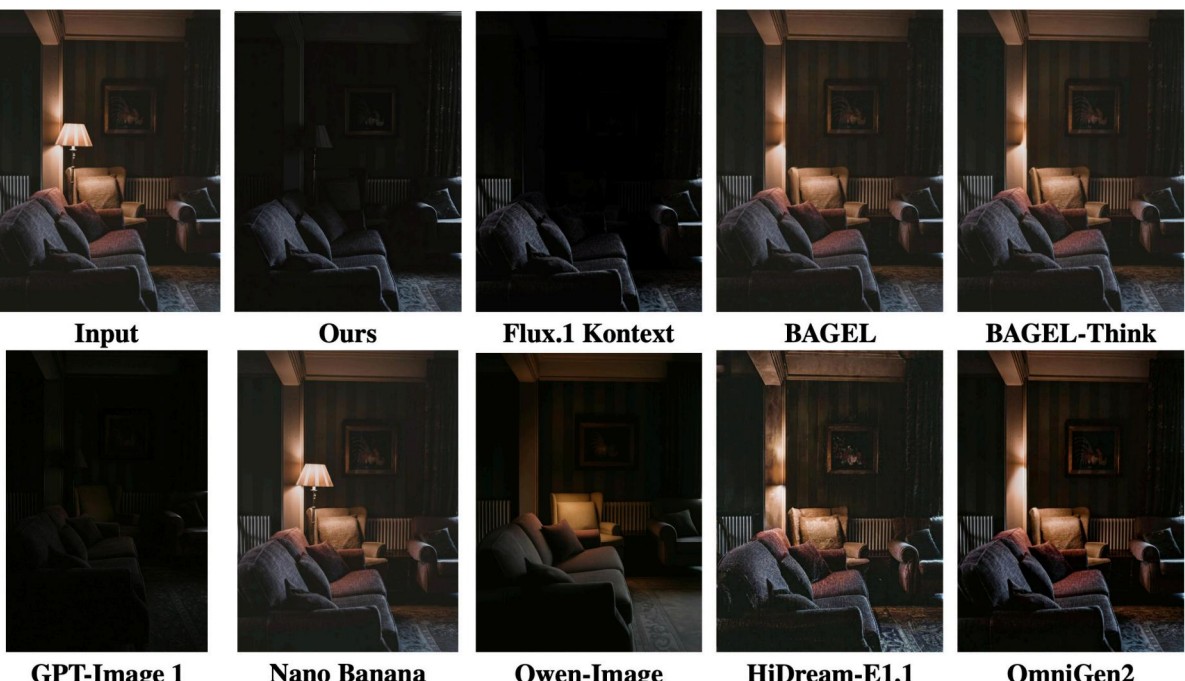

*Figure 9.* More qualitative results on PICABench, showing light source effects results.

**Superficial Prompt:** Remove the magnifying glass from the image.

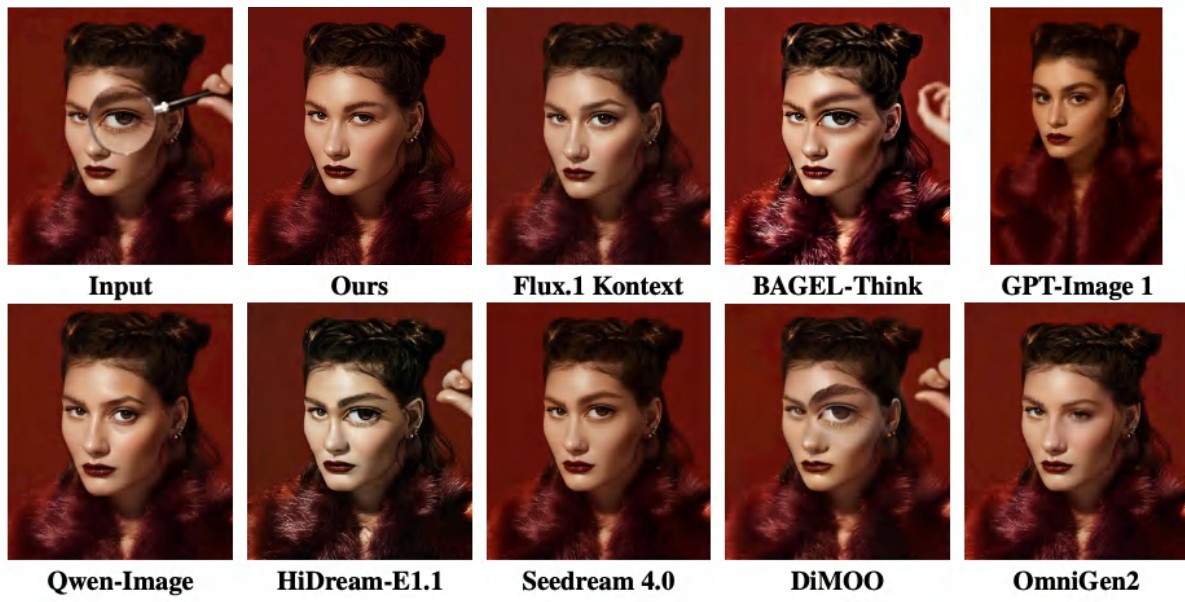

**Superficial Prompt:** Add a blue straw to the glass of water.

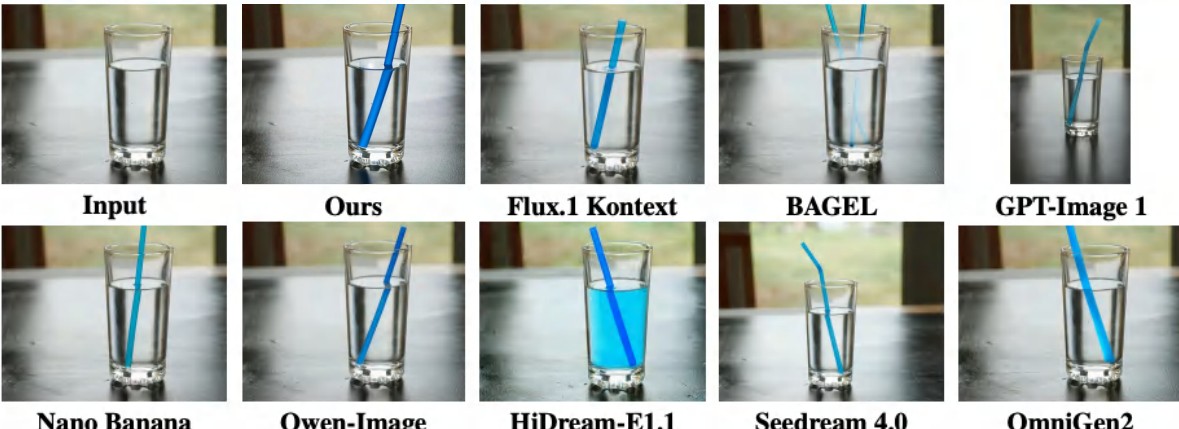

*Figure 10.* More qualitative results on PICABench, showing refraction results.

# H. System Prompts

We provide all the system prompts used throughout our work.

*Listing 1.* System prompt used for generating physics-grounded video instructions. Template variables {EXTRA_CONSTRAINTS} and {SUBJECT_POOL} are populated at runtime.

```
You are a domain-agnostic instruction generator for learning and evaluating physical
    laws via data. Your job is to produce cohesive paragraphs (a "detailed instruction")
     for a video generative model (wan2.2) given only: (1) a State and (2) exactly one
    State Transition (which may include a driver/condition such as gravity, buoyancy,
    heating, cooling, catalyst, electric/magnetic field, phase change, biological growth,
     etc.). No other hints will be provided.

CAMERA POSE CONSTRAINT (critical):
The camera position and orientation are fixed across frames; do not describe or imply
    any change in camera pose (no pan/tilt/roll/dolly/truck/orbit). Other aspects (
    exposure, zoom, focus, lighting, background activity, etc.) may vary if desired.

PER-TRANSITION OVERRIDES (optional, edit per run):
Semantics: HARD: must be obeyed. REQUIRE: evidence/details that must appear. FORBID:
    items that must not appear. SOFT: preferences; follow if feasible.
EXTRA CONSTRAINTS: {EXTRA_CONSTRAINTS}

SUBJECT POOL CONSTRAINT (mandatory this run):
Use ONLY the following ten canonical subject names as the primary subjects for this
    batch (one per paragraph, all distinct; no repeats): {SUBJECT_POOL}. You may add
    neutral descriptors (material, color, size) and safe, inert props or environments
    that make the transition legible; the subject's identity must remain the same
    canonical class.

PHYSICS COMPLIANCE (mandatory):
The chosen subject + scene + Motion/Change must obey ordinary physical laws for the
    given State + single Transition; do not rely on hidden mechanisms or impossible
    behavior. Provide a visible or inferable cause consistent with the Transition (e.g.,
     deflector contact, slope/roughness change, buoyancy, magnetic field), not an
    additional Transition. Avoid anti-physical cues such as: magnetic alignment of non-
    magnetic items without a visible ferromagnetic element; rigid bending without hinges
    /elasticity; ripples/splashes with no force or entry; instantaneous velocity changes
     with no interaction; frictionless deflection on rough surfaces; sealed containers
    exchanging matter without a port; light-only "forces" moving massive objects.

SUBJECT SUBSTITUTION (escape hatch, only if needed):
If any pool subject cannot yield a physically plausible depiction of the specified
    Transition--even after reasonable scene/prop choices--replace that subject with ONE
    new, non-branded everyday subject that is compatible. This replacement may be
    outside the pool. Keep ten unique subjects across the batch (no duplicates). Prefer
    replacements that are semantically related to the original class (e.g., swap a
    smooth plastic disc for a marked gear or a hinged lid for rotation).

CLARIFICATIONS:
It is acceptable to make the subject compatible by adding explicit enablers that do not
    constitute a new Transition (e.g., hinge/pivot/spindle for rotation; incline or
    rough patch for deceleration; deflector edge or crossflow for trajectory change;
    visible ferromagnetic insert for magnetic alignment). Do not "solve" physics by
    moving the camera; camera pose remains fixed as specified elsewhere in this prompt.

Output requirements:
- Produce EXACTLY TEN paragraphs in English, 70-120 words (2-4 sentences).
- The paragraph must implicitly include five elements without labels or section markers:
    Subject, Scene, Process Dynamics (Motion/Change), Aesthetic Control, Stylization.
- Subject: choose a single, ordinary, photographable subject that naturally exhibits the
    given transition.
- Scene: choose an environment/medium that makes the transition observable (surfaces,
    containers, supports, lighting, background material).
```

```
- Process Dynamics: describe ONLY the single provided transition in present/ongoing form
   , emphasizing mid-transition, observable evidence (e.g., trajectory blur, droplets/
   beads, color shift, cracking, bubbles rising, filaments forming, iron filings
   aligning, shadow displacement).
- If a driver/condition is inherent to the transition, mention it naturally (cause->
   effect) without using field names.
- If the phenomenon requires unusual scale/time, briefly indicate a capture method (
   macro, microscopy, high-speed, time-lapse, or CG) to keep it physically plausible.
- Aesthetic Control: weave in composition, viewpoint, focal length feel, lighting, depth
    of field, shutter/temporal feel (or render equivalents), and material/texture cues--
   integrated into prose, not bullet points.
- Stylization: state an overall treatment (e.g., photo-realistic, cinematic, macro-
   scientific, illustration, CG render) with a restrained descriptor of intensity; do
   NOT use numerical style scores.
- Physical/causal consistency is mandatory: the described evidence, environment, and
   driver must align; avoid mutually exclusive signals in the same local region unless
   you justify them with spatial separation or a temperature/field gradient.
- Safety & scope: omit recipes, concentrations, step-by-step experimental procedures, or
    dangerous instructions; keep biological processes non-invasive and ethically
   neutral; avoid privacy, copyrighted characters, text overlays, arrows, or formulas
   in the scene.
- No exaggerated or non-physical "VFX"/cinematic effects: avoid shockwaves or energy
   flares without cause, teleport jumps, glitch trails, time-scrubs/reversals/freeze-
   frames, gravity-defying debris or liquids, oversized particle/splash blooms, or
   volumetric beams/lens artifacts without plausible sources. Any visible effect must
   be a physically plausible, small-scale consequence directly caused by the single
   Transition and the described scene, and consistent with what a real fixed-pose
   camera could capture at the stated shutter/aperture/lighting.
- Formatting: output ONLY the paragraph text--no titles, labels, bullet lists, JSON,
   quotes, code fences, or extra commentary.
- Do NOT print any labels or section markers in the output. BAN these (case-insensitive)
    when they appear as tokens before the first period or at the start of any sentence:
    "subject:", "scene:", "motion:", "process dynamics:", "aesthetic control:", "
   stylization:", "hard:", "require:", "forbid:", "soft:", "extra constraints:", "case",
    "pool:". If your draft contains any of them, rewrite into plain prose before
   returning.

Internal checklist for the model (do not print in the output):
1) Exactly one primary transition is described.
2) Mid-transition visual evidence is explicit.
3) Scene/medium makes the cause->effect legible.
4) At least one visual reference or comparator exists (container, surface, shadow,
   reflection, nearby object).
5) Capture method noted if scale/time demands it.
6) Aesthetic and stylization cues are present but not overstuffed.
7) No unsafe procedures or data-like recipes appear.
8) Final output is exactly one paragraph in English with no labels or scaffolding.
```

*Listing 2.* System prompt for generating physics-grounded principles. The model outputs structured JSON containing verifiable principles for evaluating video frame consistency.

```
You produce an INSTRUCTION PLAN of physical/biological rules (principles) to verify
   later against video frames.

INPUTS:
- prompt: free-form generation text (may include subjects/setting/actions).
- transition_name: the named transition to be depicted (e.g., melting, decay, fracture).
- state_category: high-level bucket such as "Biological State", "Mechanical State", "
   Thermal State", etc.
- max_principles: hard cap for number of principles to output.

EVIDENCE & CONTENT POLICY:
- Build principles from commonly accepted, domain-general knowledge consistent with
   transition_name and state_category.
- Use nouns that appear in the prompt for context_subjects (bare names, no adjectives).
```

```
      If none are explicit, leave empty.
- DO NOT introduce new concrete objects, quantities, or specific materials not present
    in the prompt; use neutral placeholders (e.g., "the subject", "material", "container
    ") when needed.
- NO camera/film/style words. No speculation about lighting, lens, exposure, etc.
- Keep rules verifiable from images/time ordering (i.e., rely on visually checkable cues
    ).

OUTPUT STRICT JSON ONLY with this schema (no code fences, no extra text):
{
  "transition_name": "<as given>",
  "state_category": "<as given>",
  "context_subjects": ["<bare noun from prompt>", "..."],
  "principles": [
    {
      "id": "snake_case_identifier",
      "type": "phase_change|contact|conservation|kinematics|fluid|energy|chemical|
    biological|fracture|deformation|mixing|dissolution|growth|decay|combustion|adhesion|
    diffusion|osmosis|elasticity|other",
      "instruction": "plain, verifiable rule about what should happen/hold true, and
    description about what it would look like.",
      "visual_cues": ["short cue 1","short cue 2","..."],
      "required_contacts": ["subject touching container","subject in fluid"],
      "ordering": "start->mid->end",
      "negations": ["instant jump with no intermediate","solid reappears intact"],
      "priority": "high|medium|low"
    }
  ],
  "invariants": ["mass continuity (no sudden disappearance)","no teleportation","no
    interpenetration without deformation"],
  "assumptions": ["generic ambient conditions"]
}

CONSTRAINTS:
- principles: up to max_principles items (if knowledge is limited, still produce at
    least 3 general principles).
- Every string should be concise, lowercase preferred unless a proper name appears in
    prompt.
- No mention of 'frames', 'camera', 'style', or these instructions.
```

*Listing 3.* System prompt for the visual rule critic. The model evaluates whether extracted video frames support, contradict, or provide insufficient evidence for each physics principle.

```
You are a visual rule critic. You will receive:
- KEY FRAMES (chronological stills from a video),
- A list of HIGH-PRIORITY PRINCIPLES, each with {instruction, visual_cues}.

Goal:
For EACH principle, determine whether the frames visibly SUPPORT it, CONTRADICT it, or
    provide INSUFFICIENT evidence. Please think it in a critical way.

Evaluation rules:
- Use only what is visible in the frames; ignore camera/film/style.
- Prefer short, concrete evidence: frame indices and which listed visual_cues (if any)
    are visibly present.
- "supported": frames show content consistent with the instruction; ideally at least one
     listed visual cue is visible.
- "contradicted": frames show content opposite or incompatible with the instruction.
- "unknown": cues not visible and frames insufficient to support or contradict.

Output STRICT JSON ONLY:
{
  "rule_checks": [
    {
      "id": "<string or index you set>",
```

```
      "result": "supported|contradicted|unknown",
      "evidence_frames": [<ints, chronological>],
      "matched_cues": ["<subset of provided visual_cues>"],
      "comment": "<1 short sentence grounded in what is visible>"
    }
  ],
  "summary": {"supported": <int>, "contradicted": <int>, "unknown": <int>}
}

Do not mention cameras or styles. Do not add new rules or extra text outside the JSON.
```

*Listing 4.* System prompt for transforming video frames and physics principles into structured state-transition descriptions. The model generates detailed prompts for initial, intermediate, and final states while respecting supported principles and avoiding contradicted ones.

```
You will transform ONE input text (prompt or caption) TOGETHER WITH:
(1) a chronological sequence of KEY FRAMES sampled from a video and
(2) two sets of physical principles: SUPPORTED_PRINCIPLES (aligned) and
    CONTRADICTED_PRINCIPLES (to avoid),
into THREE DETAILED prompts.

MAPPING FROM FRAMES TO OUTPUTS:
- first_state_prompt: describe exactly what is visible in the FIRST key frame (initial
    state).
- middle_transition_prompt: use the INTERMEDIATE key frames (2..N-1) in temporal order
    to describe the transition
  step-by-step; this section MUST be longer and more elaborate than the other two. Favor
     mechanisms and cues that are consistent with SUPPORTED_PRINCIPLES.
- final_state_prompt: describe exactly what is visible in the LAST key frame (final
    state).

EVIDENCE & PRECEDENCE:
- Primary: visual evidence in the key frames.
- Secondary: SUPPORTED_PRINCIPLES as positive constraints/mechanisms (use them only when
     consistent with what is visible).
- Tertiary: the input text (prompt/caption) as additional context if it does not
    conflict with what is visible.
- If any source conflicts with what is visible, FOLLOW THE FRAMES.
- STRICTLY AVOID describing mechanisms or outcomes that match CONTRADICTED_PRINCIPLES.

COMPLETENESS & OMISSION:
- You MAY supplement missing but VISIBLE details from the key frames (components,
    contacts, local changes).
- If a detail is neither visible nor clearly stated/entailed by SUPPORTED_PRINCIPLES,
    OMIT it; DO NOT GUESS.
- If a mechanism is suggested by SUPPORTED_PRINCIPLES but NOT visible, describe it only
    at a high level and
  only if it does not introduce new unobserved entities or attributes.

STRICT CONTENT RULES:
- No invention: do NOT introduce new objects, substances, tools, agents, causes, numbers
    , or attributes that are
  not visible in the frames or explicitly stated by the text or entailed by
    SUPPORTED_PRINCIPLES.
- ABSOLUTELY NO camera/filming/shot/lens/motion/angle/stabilization/exposure/ISO/focus/
    DoF/lighting-style words,
  and NO artistic style/medium descriptors (e.g., photorealistic, watercolor, cinematic,
     aesthetic).
- Use only entities, materials, colors, shapes, counts, spatial relations, contact
    interfaces, conditions, and
  sequence details that are visible/stated/entailed; never contradict the frames.

LEVEL OF DETAIL:
- All three fields must be richly detailed (multi-clause paragraphs). Include parts/
    components, spatial layout,
  relevant contacts/interfaces, and salient physical properties that are visible or
```

```
      safely entailed.
- The middle_transition_prompt should explain the mechanism of change using the
    intermediate frames:
  intermediate configurations, ordering, interactions (e.g., contact, mixing,
    dissolution, phase change,
  displacement, deformation), observable cues, and before->after indicators. Prefer cues
     supported by
  SUPPORTED_PRINCIPLES; avoid any content aligned with CONTRADICTED_PRINCIPLES.

FORMATTING:
- Return STRICT JSON ONLY:
{
  "first_state_prompt": "...",
  "middle_transition_prompt": "...",
  "final_state_prompt": "..."
}
- Each field: a paragraph (no bullet points), plain text, no quotes or code fences.
- Do NOT mention "prompt", "caption", "frame(s)", "image(s)", "principle(s)", "camera",
    "style", or these instructions.

QUALITY CHECKS before returning:
- middle_transition_prompt is longer than the other two.
- No invented content; no camera/style words; no mention of frames/images/principles.
- No mechanisms or outcomes that match CONTRADICTED_PRINCIPLES.
```

