# OpenReview forum: "From Statics to Dynamics: Physics-Aware Image Editing with Latent Transition Priors"
_ICML.cc/2026/Conference — ICML 2026 regular_

### Official Review · Reviewer_vEDw · 2026-03-02

**Soundness:** 2
**Presentation:** 3
**Significance:** 2
**Originality:** 3
**Overall Recommendation:** 4
**Confidence:** 4

**Summary:**

This paper introduces PhysicEdit, a framework for image editing. Different from old methods that just map one image to another, this paper treats editing as a physical state transition. Because normal datasets do not have these physical dynamics, the paper creates a synthetic dataset called PhysicTran38K. The paper uses Wan2.2-T2V to generate them and uses ViPE and GPT-5-mini to filter the data. The paper uses a textual-visual dual-thinking mechanism based on Qwen-Image-Edit. It uses a frozen Qwen2.5-VL for text reasoning and learnable transition queries to give visual guidance. Finally, the paper joins visual features together using a special timestep-aware modulation schedule during the diffusion process.

**Compliance With Llm Reviewing Policy:**

Affirmed.

**Final Justification:**

The authors' response has strengthened my assessment of this work. I will maintain my positive score of 4.

**Key Questions For Authors:**

1. Synthetic videos are not the same as real-world physics. If the video generation model makes physical mistakes, PhysicEdit will learn these errors too. The authors need to explain why using synthetic data is enough to represent the complex physics of the real world.
2. The paper relies only on GPT-5 to evaluate physical realism in PICABench. But the data construction and filtering also use the GPT-5 model family (GPT-5-mini). The paper should add a human study or use other models to prove the evaluation is fair and not biased.

The authors must explain these issues clearly.

**Limitations:**

Yes

**Strengths And Weaknesses:**

## Strengths
1. Formulating image editing as a continuous physical state transition rather than a discrete semantic mapping is a logical and promising direction for the field.
2. It is end-to-end, requiring no video input at inference, which maintains compatibility with standard editing workflows.
3. The paper provides clear and reproducible descriptions for experimental protocols and code assets. These details are helpful for future research and help other people to reproduce the work easily.
## Weakness
1. The whole PhysicTran38K dataset is generated by a synthetic model. These video models are not good at real-world physics and often hallucinate. Learning physics from a hallucinating generator is risky. Also, the paper uses GPT-5-mini for principle-driven verification. However, LLMs can also have physical hallucinations and they only evaluate superficial visual cues from keyframes, not the true physical dynamics.
2. The paper uses a simple linear formula to mix DINO and VAE features during the diffusion process. This design is empirical and lacks a strong theoretical explanation. Even though the ablation study shows this is better than using only one feature or a hard switch method, the paper does not try more theoretical guidance mechanisms. This makes the current formulation look a bit arbitrary.
3. The paper relies heavily on automated evaluation using VLMs and LLMs. There is no human study to double-check if the physics are really correct for people. Also, the GPT-5 used for evaluation is from the same family as the GPT-5-mini used in the data construction pipeline. This overlap between the evaluator and the data generator might cause a bias and make the performance scores higher than they should be.
4. Compute and runtime overhead of the added components (reasoning + queries) compared with baselines are not reported.

---

> ### Author Rebuttal · Authors · 2026-03-29
>
> We thank Reviewer vEDw for the thorough and thoughtful review. We appreciate the reviewer’s recognition of the soundness and potential of our problem formulation, the end-to-end design, and the clear and reproducible experimental setup. Below, we address the concerns in detail.
>
> ## **W1 & Q1: Synthetic Data Reliability and Physical Hallucination**
>
> We understand the reviewer’s concern regarding the use of synthetic video data and the reliability of LLM-based filtering. We would like to clarify that PhysicEdit is not intended to learn exact physical laws or perform first-principles simulation. Instead, our goal is to learn **physics-aware transition dynamics priors**, namely, regularities in how visual scenes evolve under physical changes such as melting, refraction, or deformation. We believe that this is a weaker and more practical objective: the model is trained to capture plausible transition patterns, rather than to recover physically exact trajectories in a strict simulation sense.
>
> Regarding data reliability, we refer the reviewer to our response to **Reviewer ueFC — W2**, where we provide a detailed discussion of the dataset construction pipeline.
>
> ## **W2: Theoretical Justification for DINO/VAE Linear Mixing**
>
> We thank the reviewer for requiring deeper justification. Our design is motivated by the coarse-to-fine behavior commonly observed in the diffusion denoising process: earlier high-noise steps are more associated with global structure, while later low-noise steps focus on local details [1,2]. DINO features better capture high-level semantic and structural information, while VAE features contain more local appearance details [3]. We therefore use a simple linear schedule that places greater weight on DINO features at higher noise levels and on VAE features at lower noise levels. We do not claim that this schedule is theoretically optimal, but view it as a simple and stable design consistent with the denoising process. Specifically, Table 4 in the paper shows that it outperforms both single-feature baselines and hard-switching variants.
>
> ## **W3 & Q2: Evaluation Bias and Human Study**
>
> We fully acknowledge the concern about potential evaluator-generator overlap when using GPT-5 for evaluation alongside GPT-5-mini in data construction. To provide an independent assessment, we have conducted a human evaluation study. Please refer to our response to **Reviewer sccr — W1**.
>
> ## **W4: Compute and Runtime Overhead**
>
> We report the computational overhead of PhysicEdit compared to the Qwen-Image-Edit baseline below. All measurements are conducted on a single A100 80GB GPU for 1024×1024 image generation, with default sampling steps and seed.
>
> **R4-Table 1. Inference cost comparison.**
>
> |  | Qwen-Image-Edit | PhysicEdit |
> | --- | --- | --- |
> | Textual reasoning | — | 37s |
> | Diffusion generation | 82s | 87s |
> | Total inference time | 82s | 124s |
> | Peak GPU memory | 55.88 GB | 56.55 GB |
>
> The main overhead comes from the additional textual reasoning stage, which takes 37 s, where the frozen Qwen2.5-VL generates physically grounded reasoning for the editing instruction. By comparison, the diffusion generation time increases only slightly from **82 s** to **87 s**, indicating that incorporating the transition query tokens into the MMDiT introduces limited overhead. The increase in peak GPU memory is also small, rising from **55.88 GB** to **56.55 GB**. We view this additional runtime as moderate, given the performance gains of PhysicEdit, and reducing the cost of the reasoning stage through a more efficient reasoning module remains an important direction for future work.
>
> **References**
>
> [1] Liew, Jun Hao, et al. "Magicmix: Semantic mixing with diffusion models." *arXiv preprint arXiv:2210.16056* (2022).
>
> [2] Dieleman, Sander. "Spectral Autoregression." *Blog post*, https://sander.ai/2024/09/02/spectral-autoregression.html, 2024.
>
> [3] Zheng, Boyang et al. "Diffusion Transformers with Representation Autoencoders." *arXiv preprint arXiv:2510.11690* (2025).

---

> > ### Author Rebuttal · Reviewer_vEDw · 2026-04-02
> >
> > Thanks for your response. My concerns have been addressed, and I will keep my rating.

---

> > > ### Author Response · Authors · 2026-04-03
> > >
> > > Thank you for the update. We are glad that our responses addressed your concerns.

---

### Official Review · Reviewer_ueFC · 2026-03-09

**Soundness:** 3
**Presentation:** 3
**Significance:** 3
**Originality:** 3
**Overall Recommendation:** 4
**Confidence:** 4

**Summary:**

This paper reframes instruction-based image editing from a static source-to-target mapping into a physical state transition problem. The authors argue that current editors are semantically strong but often physically implausible, then propose a transition-centric framework trained with video-derived supervision. They introduce a new dataset (PhysicTran38K) and a model (PhysicEdit) designed to improve physical realism in edited images based on the powerful qwen-image-edit with model modification.

**Compliance With Llm Reviewing Policy:**

Affirmed.

**Final Justification:**

I will keep my positive score for my main concern has been solved.

**Key Questions For Authors:**

See the weakness part.

**Limitations:**

They have not discussed the limitation. The main limitation of this work is that their method is still not 100% correct. They only beat the baselines with minor performance improvement. All the evaluations are determined by the GPT-5, which further introduced more bias of the model.

**Strengths And Weaknesses:**

### Strengths

1. The paper addresses an important and practical problem: current image editors often produce semantically correct but physically implausible results and the core reframing from static image mapping to physical state transition is clear, novel, and well motivated.
3. The proposed method is coherent: physically grounded textual reasoning and latent visual transition priors are complementary by design.
4. The dataset contribution (PhysicTran38K) is substantial, with a structured taxonomy and explicit transition-oriented supervision.
5. The reported results show consistent gains over strong open-source baselines on both physical realism and knowledge-grounded editing benchmarks.
6. The paper is well organized, with useful ablations that support the role of key components.


### Weaknesses

1. Physical realism is evaluated mostly through benchmark/model-based scoring rather than strict objective physics metrics. Can the authors add quantitative, physics-grounded metrics (e.g., geometric/optical consistency checks) and more rigorous human-blind evaluations?

2. The training data is largely synthetic by Video Diffusion Model (Wan-14B) and LLM-filtered, which may introduce generation/annotation bias, and the generation might introduce additional (visual) quality degradtation comparing with the image model.  How to avoid this phenomenon?

3. The author introduces additional tokens (DINO Head) for generation, which might introduce additional bias. How to evaluate the effectiveness tradeoff between the editing dataset and the network structure? As we all know, the scaling of the dataset will be more important than the introduced bias of the network.

4. Does the author visualize or evaluate the tokens of the transition queries? which can be decoded by the VAE decoder to show the effectiveness of the transition reasoning?

---

> ### Author Rebuttal · Authors · 2026-03-30
>
> We thank Reviewer ueFC for the detailed review and the positive assessment of our problem formulation, method, dataset, experiments, and presentation. Below we address the concerns.
>
> ## W1: Physics-Grounded Metrics and Human Evaluation
>
> We agree that more explicit physics-grounded evaluation would be helpful. However, such metrics are difficult to define or apply in our setting, because many physically meaningful quantities, such as velocity, acceleration, are not directly achievable from a source–target image pair without access to temporal trajectories or material properties. Similarly, optical-flow-based metrics are designed for video frames, therefore not applicable to our single-image editing setting.
>
> For this reason, we adopt VLM-based evaluation as a unified protocol across diverse scenarios, and we additionally supplement it with human evaluation. Please refer to our response to **Reviewer sccr — W1**.
>
> ## W2: Synthetic Data Bias and Visual Quality Degradation
>
> We thank the reviewer for this important concern. We would like to address it in two directions.
>
> **On synthetic data reliability.** Our dataset construction uses a two-stage filtering pipeline. First, ViPE-based filtering removes videos with significant camera motion, which is important because our target task is image editing rather than video generation, and large camera changes would weaken the supervision for static image editing. Second, GPT-5-mini performs principle-driven verification, where each candidate trajectory is checked against the physical principles relevant to its transition type. Only trajectories that satisfy these criteria are retained. More broadly, although synthetic videos might not be perfect, they provide transition-level supervision that is absent from conventional static image-pair datasets. In this setting, the supervision need not be physically correct in every case to be useful; what matters is whether it captures informative transition patterns, which our experimental results suggest it does.
>
> **On visual quality degradation.** Our method uses LoRA fine-tuning on top of Qwen-Image-Edit, so the pretrained backbone remains largely frozen and the adaptation is lightweight. Practically, we do not observe degradation in general editing performance. As reported in our response to **Reviewer ZafP — W2**, PhysicEdit improves ImgEdit-Bench from 4.35 to 4.40 and GEdit-Bench G.O from 7.56 to 7.87, suggesting that the proposed adaptation preserves, and in some cases slightly improves, general editing quality.
>
> ## W3: Dataset and Network Structure Effectiveness Tradeoff
>
> To disentangle the contribution of the proposed dataset from that of our architectural design, we conduct a controlled ablation in which we fine-tune Qwen-Image-Edit on PhysicTran38K using standard LoRA-based supervised fine-tuning(SFT), under the same training schedule and number of epochs, while removing the proposed network architecture. This setting isolates the effect of the dataset alone.
>
> **R3-Table 1. Ablation: Dataset (SFT) vs. Dataset + Architecture (PhysicEdit).**
>
> | Model | PICABench | KRISBench |
> | --- | --- | --- |
> | Qwen-Image-Edit | 61.26 | 65.56 |
> | +SFT | 61.79 (+0.53) | 67.09 (+1.53) |
> | PhysicEdit  | 64.86 (+3.60) | 72.16 (+6.60) |
>
> The results suggest that the dataset alone provides only limited gains under conventional SFT, with improvements of 0.53 on PICABench and 1.53 on KRISBench over the Qwen-Image-Edit baseline. In contrast, PhysicEdit, which uses the same training data together with the proposed transition-query design and dual-head supervision, achieves much larger improvements of **3.60** and **6.60**, respectively. This comparison indicates that the benefit of PhysicTran38K is not fully realized through SFT alone; instead, the proposed architectural components play an important role in exploiting the transition information contained in the dataset.
>
> ## W4: Visualization of Transition Query Tokens
>
> We thank the reviewer for this helpful suggestion, which was also raised by Reviewer ZafP. Please refer to our response to **Reviewer ZafP — W1** for a detailed discussion.
>
> ## Limitation Discussion
>
> We thank the reviewer for pointing out the lack of a limitation discussion. In the final version, we will add a separate Limitation section to discuss several important aspects of our current work, including the reliance on synthetic video data and its potential biases, the reduced interpretability of latent transition queries, and the current scope of the evaluation.

---

> > ### Author Rebuttal · Reviewer_ueFC · 2026-04-03
> >
> > The rebuttal provides multiple experiments to address my main concerns. I will keep my rating.

---

> > > ### Author Response · Authors · 2026-04-03
> > >
> > > Thank you for the update. We are glad that our responses addressed your concerns.

---

### Official Review · Reviewer_sccr · 2026-03-13

**Soundness:** 3
**Presentation:** 3
**Significance:** 3
**Originality:** 2
**Overall Recommendation:** 4
**Confidence:** 4

**Summary:**

This paper is discussing how to improve physical realism in instruction-baed image editing models. The current image editing model failed to get physically correct image. For example, models may insert objects correctly but fail to respect physical phenomena such as refraction. This paper reframes the task as a physical state transition problem. The authors argue that models should learn the underlying dynamics that govern how scenes evolve under physical laws. They first proposed PhysicTran38K dataset, a video-based dataset containing roughly 38K transition trajectories. Then based on that, they propsed PhysicEdit framework. The first part is Physically-grounded reasoning, a qwen model. Second is implicit visual thinking module that are Learnable transition queries are trained from video trajectories to encode intermediate state dynamics. Experiments on PICABench and KRISBench show improvements in physical realism and knowledge-grounded editing compared with existing open-source models.

**Compliance With Llm Reviewing Policy:**

Affirmed.

**Final Justification:**

The human evaluation is solid, and the issue for the demo figure isssue is also understandable. I will keep my positive score.

**Key Questions For Authors:**

1. I tested many images in this paper such as the insert straw and the performance on sota model like nana banana is actually very good, unlike what you presented in the paper. Could you specify the testing model setting in this paper?

**Limitations:**

Yes, they discussed that.

**Strengths And Weaknesses:**

Strengths:

1. Dataset contribution. The PhysicTran38K dataset appears to be one of the first attempts to systematically curate physics-oriented editing transitions from video data. The hierarchical taxonomy of physical domains is thoughtful and covers a wide range of interactions.
2. The dual-stream reasoning mechanism (textual reasoning + latent transition queries) is a creative way to combine.
3. Results show consistent gains across multiple physical realism metrics, particularly in categories such as deformation, light effects, and causal transitions.

Weakness:
1. Lack of human evaluation but mainly using GPT-5. Phyiscs is very subtle and human is important here.

---

> ### Author Rebuttal · Authors · 2026-03-29
>
> We sincerely thank Reviewer sccr for the positive and encouraging review. We appreciate the reviewer’s recognition of our dataset contribution, the dual-stream reasoning design, and the consistent gains in physical realism. Below we address the concerns in turn.
>
> ## W1: Lack of Human Evaluation
>
> We agree that human evaluation is important for assessing physical plausibility. To complement the automatic evaluation, we conducted a large-scale human preference study on a subset of PICABench using the Rapidata platform.
>
> **Experimental Setup.** We compare four models: PhysicEdit, Qwen-Image-Edit, ChronoEdit, and FLUX.1-Kontext. We evaluate all six pairwise combinations, with 100 samples per pair selected from PICABench. Each comparison is rated by 7 independent annotators, who are shown the source image, the editing instruction, and two anonymized model outputs side by side, and asked: *"Which edited image better completes the editing instruction? Consider both the accuracy of the edit and physical plausibility."* The winner is determined by majority vote. In total, we collect **4,200 individual judgments** across **600 pairwise comparisons**. We then compute Elo ratings following the protocol of Chatbot arena[1], using K=24, σ=400, Gaussian-initialized ratings (μ=1000, σ₀=300), and T=100 shuffled rounds for robustness.
>
> **R2-Table 1. Elo ratings from human evaluation on PICABench.**
>
> | Rank | Model | Elo Rating |
> | --- | --- | --- |
> | 1 | PhysicEdit | 1045  |
> | 2 | Qwen-Image-Edit | 1022  |
> | 3 | ChronoEdit | 989  |
> | 4 | FLUX.1-Kontext | 915 |
>
> **R2-Table 2. Pairwise win rate matrix (row model win % against column model).**
>
> |  | PhysicEdit | Qwen-Image-Edit | ChronoEdit | FLUX.1-Kontext |
> | --- | --- | --- | --- | --- |
> | PhysicEdit | — | 52.0% | 56.0% | 72.0% |
> | Qwen-Image-Edit | 48.0% | — | 57.0% | 62.0% |
> | ChronoEdit | 44.0% | 43.0% | — | 64.0% |
> | FLUX.1-Kontext | 28.0% | 38.0% | 36.0% | — |
>
> As shown in R2-Table 1 and R2-Table 2, PhysicEdit achieves the highest Elo rating among the four models and is the only method that wins in head-to-head comparisons against all other models. Specifically, it achieves win rates of 72.0% against FLUX.1-Kontext, 56.0% against ChronoEdit, and 52.0% against Qwen-Image-Edit. The overall win rate of PhysicEdit across all 300 head-to-head comparisons is 60.0%.
>
> Overall, these human evaluation results align with the trends observed in the automatic metrics and provide additional evidence that PhysicEdit’s improvements in physical realism are reflected in human annotators.
>
> ## Q1: Baseline Testing Settings and Comparison with SOTA Models
>
> We thank the reviewer for raising this point. The proprietary models shown in our teaser figure are the versions available in the web interface at the time of our experiments, namely **GPT-Image-1.5** and **Nano Banana Pro**. As the web interface does not provide user control over inference-time hyperparameters, these models were evaluated under the platform’s default settings.
>
> The teaser is intended only as an illustrative example showing that even strong image editing models may struggle on edits involving complex physical dynamics. It is not meant to suggest that these models consistently fail in all such cases. In our experience, GPT-Image-1.5 relatively often exhibits such physically implausible results on physics-intensive edits, making similar failure cases easy to reproduce. Nano Banana Pro is generally stronger, but its behavior is still not uniformly reliable in these settings, and isolated successful examples do not necessarily reflect its overall robustness. Our point is simply that current image editing models, despite their strong general capabilities, can still behave inconsistently when physical constraints play a central role.
>
> [1] Chiang, Wei-Lin, et al. "Chatbot arena: An open platform for evaluating llms by human preference." ICML2024.

---

### Official Review · Reviewer_ZafP · 2026-03-14

**Soundness:** 3
**Presentation:** 3
**Significance:** 3
**Originality:** 3
**Overall Recommendation:** 4
**Confidence:** 4

**Summary:**

Current image editing models follow text instructions well, but often fail when edits require realistic physical changes like refraction or deformation. To solve this, the paper introduces PhysicTran38K, a large video-based dataset, and PhysicEdit, a framework that models editing as physical state transitions with both visual and reasoning guidance. Their method achieves more physically realistic and knowledge-grounded edits than prior open-source approaches.

**Compliance With Llm Reviewing Policy:**

Affirmed.

**Final Justification:**

Thanks authors for addressing my concerns. I kept my postive score

**Key Questions For Authors:**

Please see the weakness

**Limitations:**

Yes

**Strengths And Weaknesses:**

Strongness:

The paper is clearly written.

The dataset is a great contribution.

The results are promising.

The idea of implicit visual thinking is interesting and compelling.

Weakness:

There appears to be a gap between training and inference in the implicit visual thinking module. During training, the network relies on ground-truth video keyframes to extract DINO and VAE features, whereas during inference these features are replaced by learnable queries. It would be interesting to further analyze this discrepancy, for example by reconstructing the output features generated from the learnable queries through the VAE head and conducting additional evaluations on their quality.

The current results seem to be mainly limited to single-object-related editing scenarios. It would strengthen the paper to evaluate the method on more complex settings, such as multi-object editing, where physical interactions may be more challenging.

The paper is also missing human evaluation results. Since physical plausibility and editing quality are difficult to fully capture with automatic metrics alone, human studies would provide valuable additional evidence for the effectiveness of the method.

In addition, it would be interesting to investigate whether the source of the physical reasoning text prompts affects performance. For example, are there noticeable differences if these prompts are generated by stronger language models such as ChatGPT or Gemini?

---

> ### Author Rebuttal · Authors · 2026-03-29
>
> We thank Reviewer ZafP for the thoughtful and constructive review, and for recognizing the clarity of the paper, the dataset contribution, the promising results, and the implicit visual thinking design. Below we address each concern.
>
> ## W1: Training-Inference Gap
> We thank the reviewer for this insightful question.
>
> **On the representation space of transition queries.** Although the transition queries are supervised through both the DINO and VAE heads during training, they remain in the Qwen2.5-VL embedding space, since they are used as conditioning inputs to the MMDiT backbone following the original Qwen-Image-Edit design. As a result, they are not image latents in the VAE space and cannot be directly decoded by VAE for visualizations. Instead, they provide a compact conditioning representation of transition dynamics for downstream generation.
>
> **On the training–inference gap.** The key question is not whether these queries are directly decodable, but whether they approximate the target transition representations sufficiently well during training. In practice, the transition loss becomes very small after convergence, suggesting that the learned queries closely match the GT keyframe feature targets and the gap between training-time supervision and inference-time usage is limited.
>
> **On the nature of the learned transition queries.** Our design does not aim to explicitly reconstruct intermediate frames. Instead, the transition queries encode an abstract representation of intermediate physical states that captures how a scene evolves under physical change. This design is more efficient at inference time, as it avoids generating explicit reasoning steps, and it may generalize better across instances by focusing on higher-level transition regularities rather than pixel-level realizations. As the reviewer notes, this formulation is less interpretable than approaches with explicit intermediate reconstruction. We view this as an inherent trade-off of using compact latent representations. Empirically, the consistent gains across benchmarks suggest that the learned transition queries capture useful transition dynamics despite this reduced interpretability.
>
> ## W2: Multi-Object Editing Evaluation
>
> We thank the reviewer for this suggestion. To evaluate multi-object editing beyond the original submission, we conducted further experiments on three additional benchmarks covering multi-object and reasoning-intensive scenarios. We report the most relevant metrics below.
>
> **R1-Table 1. Results on additional benchmarks for multi-object and complex editing.**
>
> | Model | ImgEdit-Bench |  | GEdit-Bench-EN | RISE-Bench |  |
> | --- | --- | --- | --- | --- | --- |
> |  | Hybrid | Overall | G_O | Spatial | Overall |
> | Qwen-Image-Edit | 3.39 | 4.35 | 7.56 | 17.0 | 8.9 |
> | PhysicEdit | 3.54 | 4.40 | 7.87 | 25.0 | 18.6 |
>
> PhysicEdit improves the multi-object Hybrid category of ImgEdit-Bench from **3.39** to **3.54**, and the Spatial category of RISE-Bench from **17.0** to **25.0**. It also improves the overall scores on all three benchmarks, including **4.35 to 4.40** on ImgEdit-Bench, **7.56 to 7.87** on GEdit-Bench-EN, and **8.9 to 18.6** on RISE-Bench. These results suggest that PhysicEdit generalizes beyond single-object physics-aware editing and remains effective in more complex, reasoning-intensive scenarios.
>
> ## W3: Human Evaluation
>
> We agree that human judgment is important for assessing physical plausibility. We therefore conducted a human study; please refer to our response to **Reviewer sccr — W1**.
>
> ## W4: Effect of Different Language Models
>
> We thank the reviewer for this question regarding the textual reasoning branch. To examine this, we replace the frozen Qwen2.5-VL-7B reasoning module with GPT-5-mini at inference time, while keeping all other components unchanged.
>
> **R1-Table 2. Effect of the reasoning model in the textual reasoning branch.**
>
> | Model | PICABench | KRISBench |
> | --- | --- | --- |
> | Qwen-Image-Edit | 61.26 | 65.56 |
> | PhysicEdit (GPT-5-mini) | 62.90 | 69.91 |
> | PhysicEdit (Qwen2.5-VL-7B) | 64.86 | 72.16 |
>
> As shown in R1-Table 2, replacing Qwen2.5-VL-7B with GPT-5-mini at inference reduce the performance of 1.96 on PICABench and 2.25 on KRISBench. At the same time, the GPT-5-mini variant still outperforms the Qwen-Image-Edit baseline. This suggests that the reasoning branch is not fully plug-and-play. A plausible explanation is that the textual reasoning tokens and transition query tokens are jointly processed by the MMDiT, so the conditioning mechanism becomes adapted during training to the distribution of Qwen2.5-VL-7B reasoning outputs. Replacing the reasoning model only at inference time introduces a distribution shift, which may harm the learned interaction of textual and visual reasoning. These results suggest that using a different reasoning model would likely require joint adaptation during training.

---

### Decision · Program_Chairs · 2026-04-30

**Decision:**

Accept (regular)

**Comment:**

This paper focuses instruction-based image-editing where the requested edit involves simulating complex physical systems. To address this, they propose a large video dataset (PhysicTran38K) containing said transformations, and PhysicEdit, a framework that models editing as physical state transitions with both visual and reasoning guidance. Overall reviewers are positive about the contribution and recommend accept. Reviewers highlight the dataset contribution, the design of the method, and the comprehensive experimental study. Concerns around limitations of the findings, and additional analysis required have been addressed by the authors.